# Neural evidence that humans reuse strategies to solve new tasks

Sam Hall-McMaster [1,2]*, Momchil S. Tomov[1,3], Samuel J. Gershman[1,4‡],
Nicolas W. Schuck[2,5,6‡]

**1** Department of Psychology and Center for Brain Science, Harvard University, Cambridge,
Massachusetts, United States of America, **2** Max Planck Institute for Human Development, Berlin,
Germany, **3** Motional AD LLC, Boston, Massachusetts, United States of America, **4** Center for Brains,
Minds, and Machines, MIT, Cambridge, Massachusetts, United States of America, **5** Institute of
Psychology, University of Hamburg, Hamburg, Germany, **6** Max Planck UCL Center for Computational
Psychiatry and Ageing Research, Berlin, Germany

‡ These authors are joint senior authors on this work
* sam_hall-mcmaster@fas.harvard.edu

## Abstract

Generalization from past experience is an important feature of intelligent systems. When faced with a new task, one efficient computational approach is to evaluate solutions to earlier tasks as candidates for reuse. Consistent with this idea, we found that human participants ($n = 38$) learned optimal solutions to a set of training tasks and generalized them to novel test tasks in a reward-selective manner. This behavior was consistent with a computational process based on the successor representation known as successor features and generalized policy improvement (SF&GPI). Neither model-free perseveration or model-based control using a complete model of the environment could explain choice behavior. Decoding from functional magnetic resonance imaging data revealed that solutions from the SF&GPI algorithm were activated on test tasks in visual and prefrontal cortex. This activation had a functional connection to behavior in that stronger activation of SF&GPI solutions in visual areas was associated with increased behavioral reuse. These findings point to a possible neural implementation of an adaptive algorithm for generalization across tasks.

## Introduction

The ability to flexibly generalize from past experience to new situations is central to intelligence. Humans often excel at this [1–3], and the reasons have long intrigued cognitive neuroscientists. A recurring theme is that neural systems should store structural information extracted from past experiences [4,5]. Structural information specifies the *form* shared by solutions to a given class of problems, abstracting over the *content* that is specific to each problem. When new situations arise, this information can be retrieved and combined with situation-specific content to make decisions.

**Academic Editor:** Matthew F. S. Rushworth,
Oxford University, UNITED KINGDOM OF
GREAT BRITAIN AND NORTHERN IRELAND

**Peer Review History:** PLOS recognizes the
benefits of transparency in the peer review
process; therefore, we enable the publication
of all of the content of peer review and
author responses alongside final, published
articles. The editorial history of this article is
available here: https://doi.org/10.1371/journal.
pbio.3003174

provided the original author and source are credited.

**Data availability statement:** Data and code are openly available from https://gin.g-node.org/sam.hall-mcmaster. This includes repositories with behavioral and magnetic resonance imaging data (sfgpi-behavioral-data, sfgpi-bids, sfgpi-fmriprep), code to reproduce all results (sfgpi-behavioral-analysis, sfgpi-neural-analysis), and additional tools (sfgpi-task, sfgpi-mriqc). Code to reproduce the results is also archived at https://doi.org/10.12751/g-node.x9fxqi and https://doi.org/10.12751/g-node.d5t523. Data and code resources are itemized in the Methods.

**Funding:** This research was supported by an Alexander von Humboldt Fellowship (https://www.humboldt-foundation.de) and a Philip Wrightson Fellowship (https://neurological.org.nz) awarded to SHM, an Independent Max Planck Research Group grant and a Starting Grant from the European Union (ERC-2019-StG REPLAY-852669, https://erc.europa.eu) awarded to NWS, and Multidisciplinary University Research Initiative (MURI) awards by the Army Research Office (W911NF-21-1-0328 and W911NF-23-1-0277, https://arl.devcom.army.mil/who-we-are/aro) to SJG. The funders had no role in study design, data collection and analysis, decision to publish, or preparation of the manuscript.

**Competing interests:** The authors have declared that no competing interests exist.

**Abbreviations:** CSF, cerebrospinal fluid; DLPFC, dorsolateral prefrontal cortex; DVARS, derivative of the root mean squared variance over voxels; EPI, echo-planar imaging; FA, flip angle; FD, framewise displacement; fMRI, functional magnetic resonance imaging; GM, gray-matter; GPI, generalized policy improvement; INU, intensity non-uniformity; ITI, inter-trial interval; MB, model-based; MTL, medial temporal lobe; OFC, orbitofrontal cortex; OTC, occipitotemporal cortex; RL, reinforcement learning; ROI, region of interest; SF&GPI, successor features and generalized policy improvement; TE, echo time; TR, measurement volume; UVFA, Universal Value Function Approximator; WM, white-matter.

Although this approach is maximally flexible, humans often use simpler forms of generalization [6]. Recent data have shown that humans tend to solve new tasks by reusing solutions from earlier tasks [7]. This relies on an algorithm known as "generalized policy improvement" (GPI; [8–10]). GPI achieves efficient generalization by storing a set of solutions (policies) that can be evaluated and selected for reuse. Importantly, a feature-based generalization of the successor representation ("successor features" or SFs) can be harnessed to support the identification of the optimal policy among those stored in memory. The resulting algorithm, successor features and generalized policy improvement (SF&GPI), is able to efficiently solve new tasks [8–10] and predict human generalization behavior [7], when tasks are situated in a common environment and each task is associated with a distinct reward function.

Here, we investigated whether the human brain implements this flexible and efficient form of generalization. If people generalize their past experiences using SF&GPI, it should be possible to detect its components in their brain activity. We developed three neural predictions based on this premise. First, we predicted that successful past policies would be represented in brain activity when people are exposed to new tasks. Second, we predicted that these policies would be prioritized, showing stronger activation than unsuccessful past policies. Third, we predicted a corresponding representation of the features associated with successful past policies, as these are used in the model to compute expected rewards.

Past research from cognitive neuroscience provides important clues about where these predictions should be observed. The dorsolateral prefrontal cortex (DLPFC) has been proposed as a region that encodes policies [11,12] and supports context-dependent action [13–17]. Based on this literature, DLPFC is a candidate region in which the encoding of successful past policies could be detected when people are exposed to new situations. The medial temporal lobe (MTL) and orbitofrontal cortex (OFC) have been proposed as regions that encode predictive representations about future states [18–22]. Based on this literature, MTL and OFC are candidate regions in which features associated with successful past policies might be detected.

To test these predictions, participants completed a multi-task learning experiment during functional magnetic resonance imaging (fMRI). The experiment included training tasks that participants could use to learn about their environment, and test tasks to probe their generalization strategy. Different reward functions were used to define different tasks in the experiment. To summarize the main results, we found that participants learned optimal solutions (policies) to the training tasks, and generalized them to test tasks in a reward-selective manner. Participant choices at test were more similar to an SF&GPI algorithm than an MB algorithm with full knowledge of the environment. Participant choices were also distinct from a model-free process. Neural results showed that optimal solutions from the training tasks could be decoded above chance during test tasks in DLPFC and (surprisingly) in occipitotemporal cortex (OTC). These solutions were also prioritized. Decoding evidence for the optimal training solutions at test was higher than alternative solutions that promised larger rewards. These results provide new insights into how a sophisticated policy reuse algorithm might be implemented in the brain.

## Results

To test whether SF&GPI computations were evident in brain activity, participants completed a gem collector game inside an MRI scanner (Fig 1). The cover story was that a criminal mastermind had hidden rare gems in different cities around the world. Participants needed to retrieve the gems to sell them for as much profit as possible. The first screen on each trial showed a set of market values that indicated the profits or losses associated with reselling each gem. Market prices could range from $2 per gem to −$2 per gem. After seeing the market values, participants faced a choice between four cities (Sydney, Tokyo, New York, London) shown in a random order on screen. Each city contained the same three gem stones (triangle, square, circle), but in different amounts. The profit earned for a particular choice was based on the combination of market values and the recovered gems. The selling price of each gem stone was first multiplied by its respective number in the chosen city, and rewards were then summed across gem stones to arrive at the total profit. Embedded in this trial structure were two important abstract elements. First, market values were reward functions that defined the task participants needed to solve on a given trial. If triangular gems had a high market value, for example, and all other factors were equal, participants would need to locate the city with the most triangular gems. Second, the gem numbers in each city defined the state features for that city, information that could be reused to guide decisions on new tasks.

Trials were ordered in a specific way to test the SF&GPI model. When beginning each block, participants did not know how many gems were present in each city and needed to learn this information by making decisions and observing the

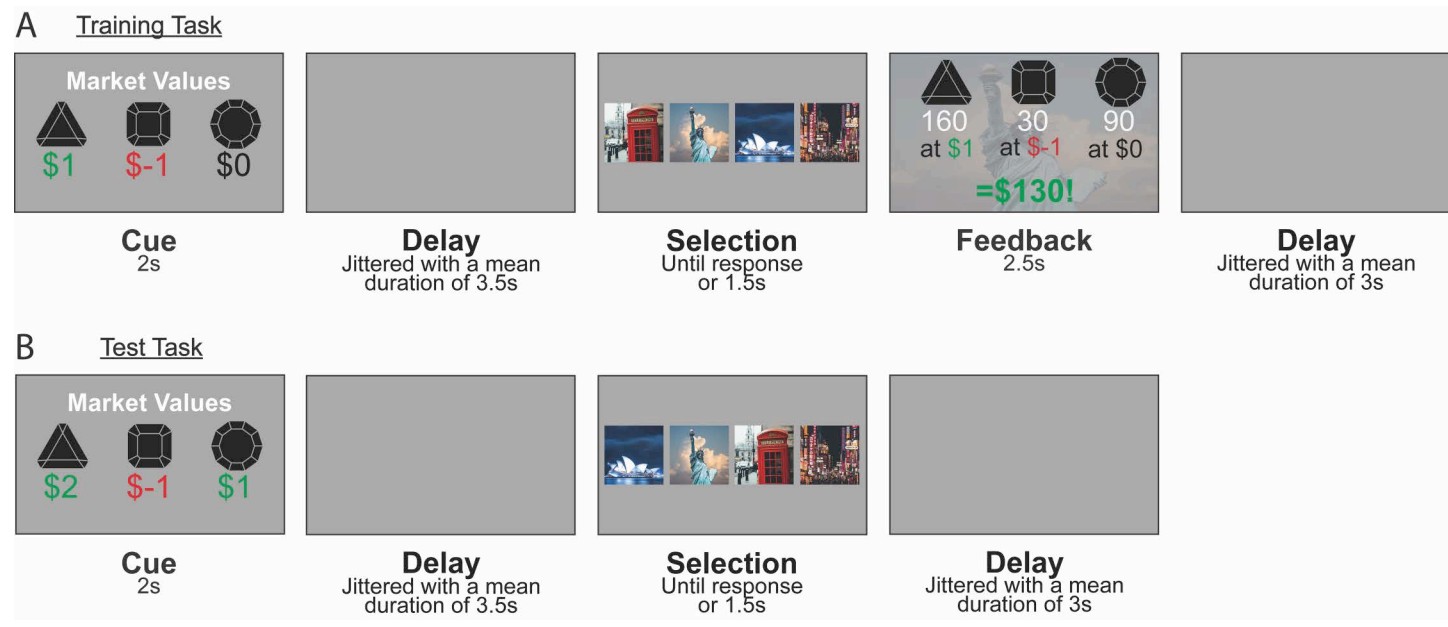

**Fig 1. Experimental design.** Participants competed a gem collector game while their brain activity was measured with functional magnetic resonance imaging. On each trial in the experiment, gems with distinct shapes could be resold for either a gain or a loss. Participants made a choice between four cities from around the world, each leading to a distinct collection of gems. To maximize profit overall, participants needed to choose the city best suited to the selling prices shown on each trial. Each block consisted of 32 training trials that included feedback (shown in **A**), followed by a mixture of 16 training trials with feedback and 20 test trials without feedback (shown in **B**). The gem collection associated with each city changed from block to block. **A**: An example training task. Following the presentation of the market values, participants selected a city and saw a feedback screen. The feedback screen revealed the gem numbers in the selected city and the profit earned on the current trial. Four training tasks were used in the experiment and were designed so that two of the four cities were optimal across training tasks. **B**: An example test task. On test tasks, participants saw a set of market values that had not appeared during training and selected a city, but did not see the outcome of their decision. Four test tasks were used in the experiment and were designed so that the two cities that were previously suboptimal now offered the highest returns. A model-based agent with a complete model of the environment is expected to be sensitive to this change. A "memory-based" SF&GPI agent that evaluates earlier task solutions as candidates for generalization is expected to choose among the cities that were optimal during training.

outcomes. This was possible during the first 32 trials of the block, which included feedback after each choice (Fig 1A). We refer to trials with feedback as *training trials* hereafter. Across training trials, participants encountered four training tasks that each had a unique reward function (i.e., a unique market value cue shown at the beginning of the trial). Two training tasks resulted in high rewards when city A was selected (e.g., Sydney), while the alternative cities resulted in losses or marginal reward. The other two training tasks resulted in high rewards when city B was selected (e.g., Tokyo), but losses or marginal reward when the alternative cities were selected. This meant that, in effect, participants needed to learn two optimal policies to perform well on training trials. One optimal policy could be used for two training tasks, and another could be used for the remaining two training tasks.

Following the initial learning phase, participants were given 20 *test trials* (Fig 1B) interspersed with 16 training trials. The test trials differed from training trials in that they introduced four new tasks (market value cues) and did not provide feedback after each choice. The test trials were designed so that the two cities, resulting in losses or marginal reward during training (e.g., New York and London), were now the most rewarding. This experimental setup allowed us to dissociate two main generalization strategies. A model-based agent with full knowledge of the environment would compute expected rewards for all four cities and choose the one that was objectively most rewarding. A model-based agent would therefore enact different policies on the test tasks compared to the training tasks. In contrast, an SF&GPI agent that stores and evaluates the best cities from training would compute expected rewards based on the optimal training policies. An SF&GPI agent would therefore choose the more rewarding city among the optimal training policies for each test task, but would not enact polices that had been unrewarding during training.

Participants completed six blocks of 68 trials in total. Training tasks had the following market values: $w_{train}$ = {[1, −1, 0], [−1, 1, 0], [1, −2, 0], [−2, 1, 0]}. Test tasks had the following market values: $w_{test}$ = {[2, −1, −1], [−1, 1, 1], [1, −1, 1], [1, 1, −1]}. Gem numbers (state features) had the following values: $\phi$ = {[120, 50, 110], [90, 80, 190], [140, 150, 40], [60, 200, 20]}. Reward for each combination of w and $\phi$ is presented in the methods (Fig 4). The vector elements in each triplet were shuffled using a shared rule before a new block (e.g., all vector elements were reordered to positions [1, 3, 2]). The mapping between cities and features (gem triplets) was also changed. These changes created the appearance of new training and test tasks in each block while preserving the structure of the experiment, in which two of the four options always proved to be best in training trials but suboptimal in test trials. Performance was incentivised with a monetary bonus that was based on the total profit accrued over all trials in the scanner game. Before the experiment took place, participants were informed about how market values and gem numbers were combined to calculate the profit on each trial. Participants also completed 80 training trials with different state features and a different task theme in preparation for the session, and 20 training trials with different state features prior to scanning.

## Participants learned optimal policies for training tasks

We first examined performance on the training tasks (Fig 2A–2E). This included training trials from the initial learning phase and training trials that were interspersed between test trials. The average reward earned per choice was higher than what would be expected if participants were choosing at random ($M = 49.77$, $SD = 10.61$, $M_{random} = −54.38$, $SD_{random} = 6.16$, $t(74) = 52.34$, corrected $p < 0.001$) suggesting successful learning. To understand training performance in more detail, we examined which choice options were selected. Training tasks were designed so that the most rewarding choice on each trial would lead to gem triplet $\phi(1)$ or gem triplet $\phi(4)$. Gem triplets are called feature triplets hereafter. Consistent with the training structure, cities associated with $\phi(1)$ and $\phi(4)$ were chosen significantly more often than the other cities (Fig 2C). $\phi(1)$ was reached more often than $\phi(2)$ or $\phi(3)$ ($M_{\phi(1)}$ = 41.66% of training trials, $SD_{\phi(1)}$ = 3.44, $M_{\phi(2)}$ = 6.52%, $SD_{\phi(2)}$ = 2.70, $t(37) = 37.78$, corrected $p < 0.001$; $M_{\phi(3)}$ = 6.38%, $SD_{\phi(3)}$ = 2.71, $t(37) = 37.95$, corrected $p < 0.001$). $\phi(4)$ was similarly reached more often than $\phi(2)$ or $\phi(3)$ ($M_{\phi(4)}$ = 45.28% of training trials, $SD_{\phi(4)}$ = 2.02, $t_{\phi(4) \text{ vs. } \phi(2)}(37) = 45.98$, corrected $p < 0.001$; $t_{\phi(4) \text{ vs. } \phi(3)}(37) = 45.42$, corrected $p < 0.001$). Choices leading to $\phi(1)$ and $\phi(4)$ were more comparable in number. However, $\phi(4)$ was reached more often than $\phi(1)$ ($t(37) = 5.34$, corrected $p < 0.001$). To understand the optimality

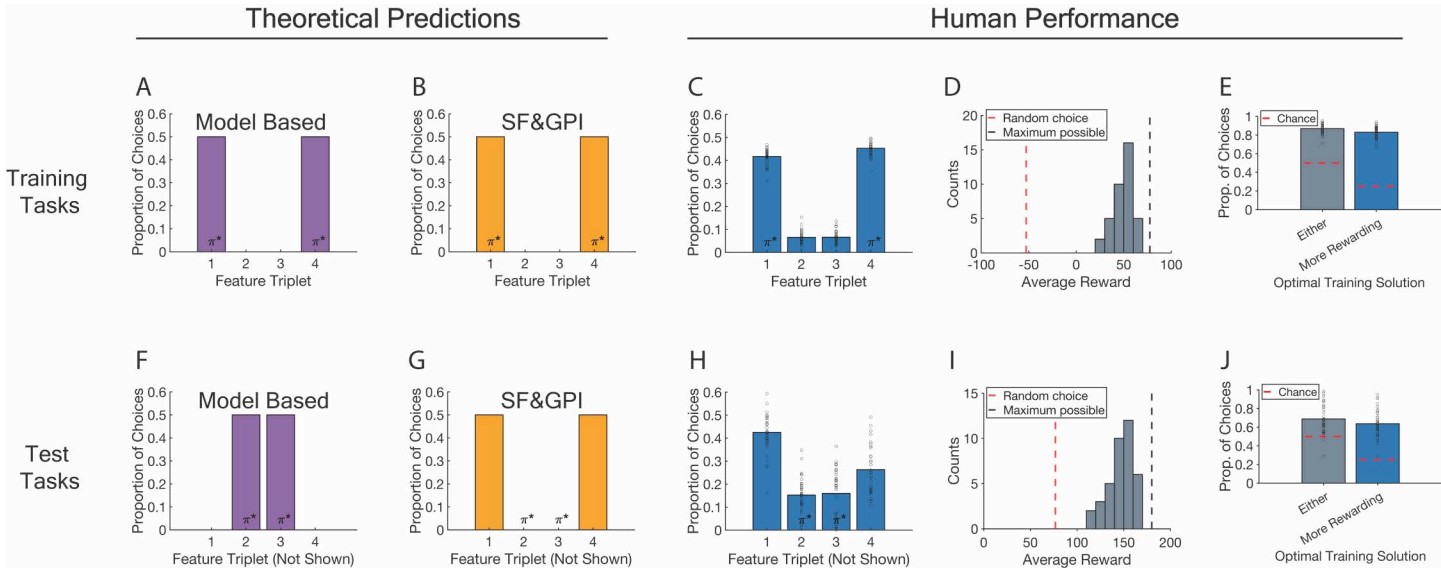

**Fig 2. Behavioral results.** Columns are grouped into theoretical predictions and human performance. Rows are grouped into training and test tasks. **A**, **B**, **F**, **G**: Theoretical predictions are shown for a model-based algorithm on training (A) and test tasks (F), as well as an SF&GPI algorithm on training (B) and test tasks (G). Each theoretical plot shows the predicted proportion of choices leading to each feature (gem) triplet. Theoretical plots show expected choice profiles after model convergence on training tasks and do not include choices during learning. **C**, **H**: Human choice profiles for training (C) and test tasks (H). The axis structure matches the theoretical plots. **D**, **I**: Performance histograms. The x-axis shows the average reward earned per trial within 10-point bins, and the y-axis shows the number of participants with this average. The dashed red line indicates the performance expected from random choices. The dashed gray line indicates the maximum performance possible. **E–J**: Use of the optimal training solutions (policies) on training tasks (E) and their reuse on test tasks (J). Both plots show the proportion of choices in which either optimal training policy was used (left bars) and the proportion of choices in which the more rewarding one was used (right bars). Dashed red lines indicate chance. A, B, C, F, G, H: π* denotes feature triplets associated with optimal training and test policies. Optimal policies for training tasks lead to feature triplet $\phi(1)$ or $\phi(4)$, and optimal policies for test tasks lead to $\phi(2)$ or $\phi(3)$. C, H, E, J: Dots show data points from individual participants. A–J: Materials to reproduce these panels are available at https://gin.g-node.org/sam.hall-mcmaster/sfgpi-behavioural-analysis.

of these decision patterns, we examined how often participants made the optimal choice on training trials. The percentage of optimal choices was significantly above chance ($M = 82.94\%$, SD = 6.36, chance = 25%, $t(37) = 56.16$, corrected $p < 0.001$, Fig 2E) indicating that participants acquired the optimal training policies. Together, these results indicate that participants acquired robust and effective decision strategies to maximize reward on the training tasks.

## Participants transferred optimal training policies to test tasks

Having shown that participants learned the optimal training policies, we turned our attention to the test tasks (Fig 2F–2J). The test tasks were designed so that choices leading to feature triplets $\phi(2)$ or $\phi(3)$ were the most rewarding. A model-based agent was expected to compute anticipated rewards under all available policies and therefore make different choices on test tasks compared to the training tasks. An SF&GPI agent that stores information about the best cities from training was expected to compute anticipated rewards only under the optimal training policies. This would result in continued choices to reach $\phi(1)$ and $\phi(4)$, rather than switching to $\phi(2)$ and $\phi(3)$. Although decisions for $\phi(1)$ and $\phi(4)$ were suboptimal, we expected that participants would still select the option that generated a higher reward among this set.

   Consistent with an SF&GPI algorithm, participants continued using optimal policies from training on most test trials ($M = 68.81\%$, SD = 15.58, chance = 50%, $t(37) = 7.44$, $p < 0.001$, Fig 2J). The choice profile on test trials was also similar to the profile seen on training trials (Fig 2H). The city associated with feature triplet $\phi(1)$ was selected significantly more often than cities that led to $\phi(2)$ and $\phi(3)$ ($M_{\phi(1)} = 42.52\%$ of test trials, $SD_{\phi(1)} = 8.27$, $M_{\phi(2)} = 15.23\%$, $SD_{\phi(2)} = 7.43$, $t(37) = 11.48$, corrected $p < 0.001$; $M_{\phi(3)} = 15.96\%$, $SD_{\phi(3)} = 10.23$, $t(37) = 9.78$, corrected $p < 0.001$). The city associated

with feature triplet $\phi(4)$ was similarly selected significantly more often than those associated with $\phi(2)$ and $\phi(3)$ ($M_{\phi(4)}$ = 26.29% of test trials, $SD_{\phi(4)}$ = 10.64, $t_{\phi(4) \text{ vs. } \phi(2)}(37)$ = 4.14, corrected $p < 0.001$; $t_{\phi(4) \text{ vs. } \phi(3)}(37)$ = 3.17, corrected $p = 0.006$). $\phi(1)$ was reached more often than $\phi(4)$ during test tasks ($t(37)$ = 9.12, corrected $p < 0.001$). While most choices were technically suboptimal, participants still performed well on test tasks overall. The average reward per choice was significantly higher than the reward expected from random choice ($M = 147.00$, $SD = 13.94$, $M_{random} = 76.89$, $SD_{random} = 7.84$, $t(74)$ = 27.02, $p < 0.001$, Fig 2I). This was due to participants selecting the more rewarding solution among the optimal training policies on test tasks significantly more often than chance ($M = 63.71\%$, $SD = 14.22$, chance = 0.25, $t(37) = 16.78$, corrected $p < 0.001$, Fig 2J). On trials where participants used either of the two optimal training policies, the more rewarding one was indeed selected in most cases ($M = 92.86\%$, $SD = 5.21$, chance = 0.5, $t(37)$ = 50.74, corrected $p < 0.001$). The proportion of SF&GPI-predicted choices on test tasks was positively related to the proportion of optimal choices during training but not to a significant degree (Spearman's $Rho = 0.294$, corrected $p = 0.073$).

Auxiliary behavioral results are presented in the supplementary information. Training performance reached an asymptote before test tasks were introduced (S1 Fig), and participants could be separated into subgroups based on how closely their behavior matched the SF&GPI predictions (S2 Fig). The SF&GPI-predicted choice was made significantly more often than the alternatives on 3 of the 4 individual test tasks (S3 Fig), with the remaining test task showing an even split between SF&GPI and MB choice, an interesting behavioral signature of Universal Value Function Approximation (S4 Fig). Generalization performance could not be explained by a simple (model-free) perseveration process (S5 Fig). The experiment also included an assessment of participants' explicit knowledge. Following scanning, participants estimated the number of gems that were present in each city during the final block of the experiment (S6 Fig). While we did not detect a difference in the mean gem estimate error across policies (corrected $p$-values > 0.09, S6A Fig), we observed a significant positive correlation between the estimate error for cities that were suboptimal during training and SF&GPI-consistent choices on test trials (Spearman's $Rho = 0.493$, corrected $p = 0.016$, S6B Fig). This suggests a possible link between noisy memory for feature triplets associated with the suboptimal training policies and the reuse of successful past solutions.

Taken together, the results in this section indicate that participants were choosing among the optimal training policies on test trials in a reward-sensitive manner, a behavioral profile more consistent with the predictions of an SF&GPI algorithm than an MB algorithm operating on a full model of the environment.

## Optimal training policies could be decoded during test tasks

Having established that behavior was consistent with the predictions of an SF&GPI model, we tested its neural predictions on test tasks: (1) that optimal training policies would be activated as decision candidates; (2) that their activation strength would be higher than alternative policies; and (3) that features associated with the optimal training policies would also be represented. We tested these predictions in four brain regions. Predictions 1–2 were expected in DLPFC due to its proposed role in policy encoding [11,12] and context-dependent action [13–17]. Prediction 3 was expected in MTL and OFC due to research implicating these regions in coding predictive information about future states [18–22]. OTC was examined as a final region due to its central role in early fMRI decoding studies and its continued inclusion in recent ones [20,23–25]. We first focus on policy activation (predictions 1–2).

To examine policy activation, decoders based on logistic regression were trained to distinguish the four cities seen during feedback on the training tasks (i.e., the cities served as training labels). One measurement volume (TR) per eligible trial was used as input for decoder training, taken 4–6 s after feedback onset to account for the hemodynamic delay. Decoders were trained separately for each region of interest (ROI). The specific time lag used for each ROI was determined in validation analyses of the training tasks, independent from our predictions about test task activity (see S7 Fig). Validation analyses further showed that decoder training locked to feedback onset on training tasks was more effective than decoder training locked to task cue onset (see S8 Fig). The decoders were applied to each TR on held-out test trials in a leave-one-run-out cross-validation procedure. This resulted in a decoding probability time course for each city on each test trial, which

reflected the evidence that a particular city stimulus was encoded in the fMRI signal. To account for class imbalances in the decoder training set, we repeated the analysis 100 times using random subsampling that matched the number of trials per category ($M = 53$ trials per training class, SD = 6.48, minimum = 37, maximum = 65). Decoding probabilities based on the cities were then coded into the following categories: (1) the more rewarding policy (among the optimal training policies); (2) the less rewarding policy (among the optimal training policies); (3) the objective best policy; and (4) the remaining policy.

We first tested the prediction that optimal training policies would be activated on test tasks, by comparing decoding evidence for the optimal training policies against chance in each ROI (Fig 3B). Test trials lasted 10 s on average (Fig 1B). Decoding evidence was therefore averaged from +2.5 s to +12.5 s following test trial onset to account for the hemodynamic delay. This revealed significant average decoding evidence for the more rewarding training policy during test tasks in OTC and DLPFC (OTC: $M = 28.07\%$, SD = 2.17, $t(37) = 8.57$, corrected $p < 0.001$; DLPFC: $M = 25.85\%$, SD = 1.51, $t(37) = 3.40$, corrected $p = 0.011$), but not in MTL or OFC (MTL: $M = 25.17\%$, SD = 1.43, $t(37) = 0.72$, corrected $p = 0.744$; OFC: $M = 25.40\%$, SD = 1.37, $t(37) = 1.79$, corrected $p = 0.405$). Numerical evidence for the less rewarding training policy was also detected in OTC, but this did not survive correction ($M = 25.97\%$, SD = 2.34, $t(37) = 2.53$, uncorrected $p = 0.016$, Bonferroni–Holm corrected $p = 0.094$, see the section entitled Reactivation and stimulus processing contributed to the decoding effects for evidence from cluster-based permutation tests). Average decoding evidence for the less rewarding training policy was not detected in the remaining ROIs (MTL: $M = 25.32\%$, SD = 1.76, $t(37) = 1.12$, corrected $p = 0.744$; OFC: $M = 25.31\%$, SD = 1.40, $t(37) = 1.38$, corrected $p = 0.708$; DLPFC: $M = 25.30\%$, SD = 1.53, $t(37) = 1.17$, corrected $p = 0.744$). Decoding evidence on individual test tasks is included in the supplement (S9 Fig). These results indicate that the more rewarding among the optimal training policies was activated in OTC and DLPFC during test tasks.

## Optimal training policies were prioritized on test tasks

Having established that the optimal training policies were activated on test tasks (prediction 1), we next tested whether their activation strength was higher than the other policies (prediction 2, Fig 3C). Average decoding evidence during the test tasks was significantly higher for the more rewarding training policy than the objective best policy in OTC, OFC, and DLPFC (OTC: $M_{diff} = 4.09\%$, $SD_{diff} = 3.69$, $t(37) = 6.75$, corrected $p < 0.001$; MTL: $M_{diff} = 0.48\%$, $SD_{diff} = 2.46$, $t(38) = 1.17$, corrected $p = 0.743$; OFC: $M_{diff} = 0.97\%$, $SD_{diff} = 1.87$, $t(38) = 3.14$, corrected $p = 0.030$; DLPFC: $M_{diff} = 1.61\%$, $SD_{diff} = 2.59$, $t(38) = 3.77$, corrected $p = 0.006$). Similar results were observed for the less rewarding training policy. Average decoding evidence for the less rewarding training policy was significantly higher than the objective best policy in OTC and DLPFC (OTC: $M_{diff} = 2.00\%$, $SD_{diff} = 3.95$, $t(37) = 3.07$, corrected $p = 0.032$; DLPFC: $M_{diff} = 1.05\%$, $SD_{diff} = 2.15$, $t(37) = 2.99$, corrected $p = 0.035$; MTL: $M_{diff} = 0.63\%$, $SD_{diff} = 2.80$, $t(37) = 1.36$, corrected $p = 0.722$; OFC: $M_{diff} = 0.88\%$, $SD_{diff} = 1.97$, $t(37) = 2.71$, corrected $p = 0.060$). Relative decoding scores above take the form: (evidence for training policy X) − (evidence for the objective best policy) and measure the extent to which optimal training policies were prioritized in neural activity. We found that relative decoding scores were significantly higher for the more rewarding training policy than the less rewarding one in OTC ($M_{diff} = 2.09\%$, $SD_{diff} = 2.37$, $t(37) = 5.38$, corrected $p < 0.001$) but comparable in other ROIs (MTL: $M_{diff} = -0.15\%$, $SD_{diff} = 1.61$, $t(37) = -0.58$, corrected $p > 0.99$; OFC: $M_{diff} = 0.09\%$, $SD_{diff} = 1.93$, $t(37) = 0.28$, corrected $p > 0.99$; DLPFC: $M_{diff} = 0.55\%$, $SD_{diff} = 1.94$, $t(37) = 1.73$, corrected $p = 0.092$). The results from testing predictions 1 and 2 suggest that successful training policies were activated and prioritized on test tasks in OTC. The more rewarding training policy was activated and prioritized in DLPFC (Fig 3B–3E). Remaining effects did not show consistent evidence when testing both predictions and are not considered further.

## Reactivation and stimulus processing contributed to the decoding effects

We next used cluster-based permutation testing [26] to estimate when the optimal training policies were activated in the decoding time courses (window tested = 2.5–12.5 s post cue; 10,000 permutations). These tests revealed significant decoding evidence for the more rewarding training policy from 5 to 12.5 s on test trials in OTC (corrected $p < 0.001$) and

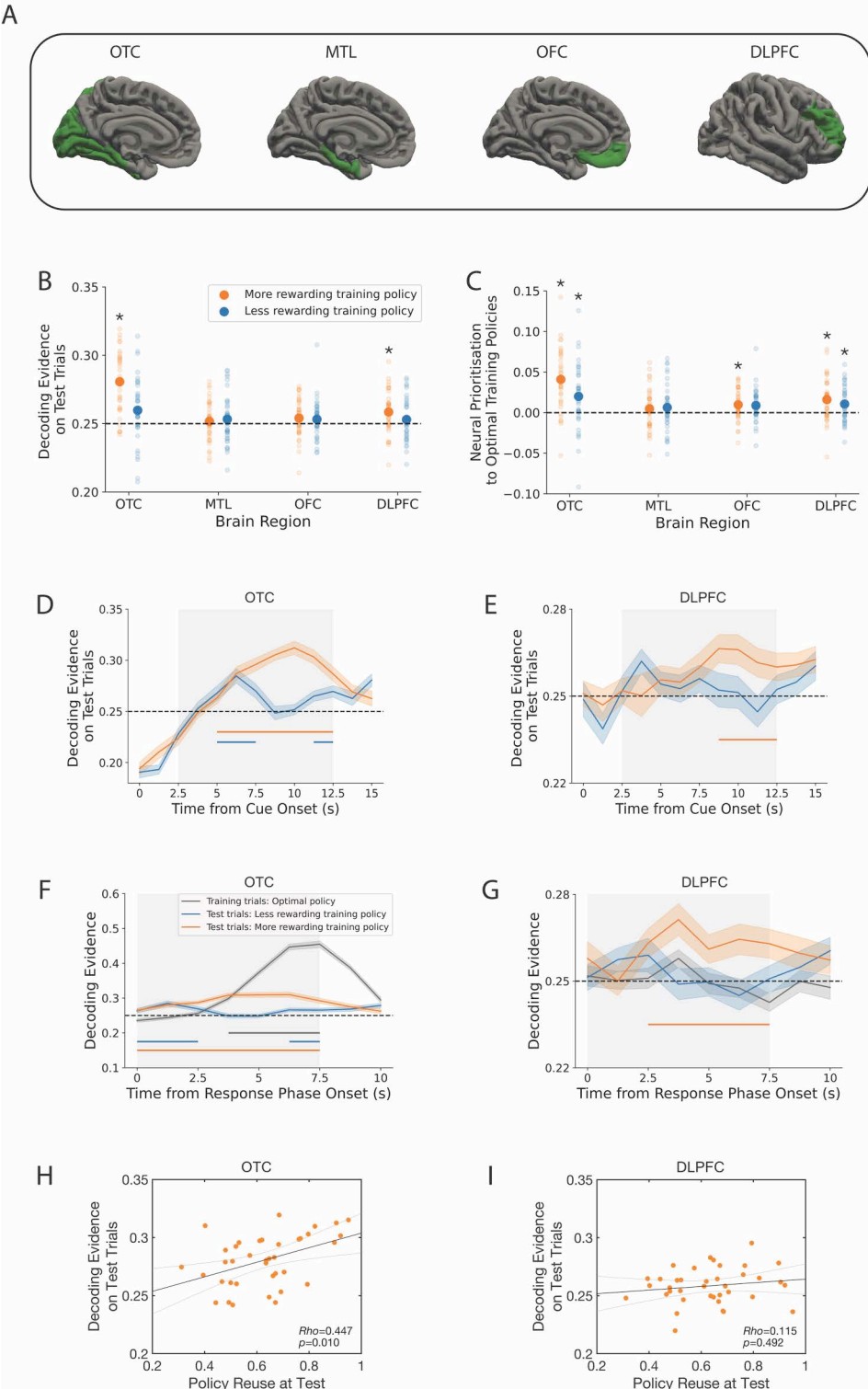

**Fig 3. Decoding results. A**: Brain regions of interest. The four regions include occipitotemporal cortex (OTC), the medial temporal lobe (MTL), orbitofrontal cortex (OFC) and dorsolateral prefrontal cortex (DLPFC). Regions were defined using FreeSurfer [27]. **B**: Decoding evidence for the optimal training policies on test tasks (y-axis) shown for each brain region (x-axis). The dashed line indicates chance. **C**: Neural prioritization to the optimal training policies. This panel shows the difference in decoding evidence on test tasks between the optimal training policies and the objective best policy.

Higher values indicate stronger evidence for the optimal training policies within the neural signal. **D**, **E**: Time resolved decoding evidence for the optimal training policies on test trials. Panels begin at the onset of the test cue. The initial period shows negative decoding evidence due to a control procedure. To avoid activity from the previous trial biasing our assessment of the current trial, we excluded trials in which the current policy category was selected on the previous trial. B–E: The effects with consistent numerical and statistical evidence across panels B–E are the activation and prioritization of the optimal training policies in OTC, as well as the activation and prioritization of the more rewarding training policy in DLPFC. The remaining effects in B–E did not show consistent evidence across analyses and are not interpreted further. **F**, **G**: Time-resolved decoding evidence locked to the response phase onset. Evidence is presented for both the optimal policy on training trials and the optimal training policies on test trials. D–G: Colored bars below each line indicate significant decoding clusters. Shaded error bars show the standard error of mean. **H**, **I**: Relationships between decoding evidence for the more rewarding training policy in a specific ROI (y-axis) and the proportion of test trials in which participants reused that policy (x-axis). Black lines indicate linear fits to the data and gray lines indicate 95% confidence intervals of the fits. B–I: Throughout the figure, decoding evidence for the more rewarding among the two optimal training policies is shown in orange, and the less rewarding among the two optimal training policies is shown in blue. The policy within this set that is more or less rewarding varies across test trials depending on the specific cue. B–G: Materials to reproduce these panels are available at https://gin.g-node.org/sam.hall-mcmaster/sfgpi-neural-analysis. H, I: Materials to reproduce these panels are available at https://gin.g-node.org/sam.hall-mcmaster/sfgpi-behavioural-analysis.

from 8.75 to 12.5 s in DLPFC (corrected $p < 0.001$). Cluster-based permutation testing is potentially more sensitive than testing average evidence because activation that occurs in smaller subsets of time points can be identified. Consistent with this idea, cluster tests identified significant decoding evidence for the less rewarding training policy in OTC from 5 to 7.5 s (corrected $p < 0.001$) and 11.25 to 12.5 s (corrected $p = 0.016$), as well as a candidate cluster at 3.75 s in DLPFC (corrected $p = 0.072$). No signfiicant decoding clusters were detected for OFC or MTL (all corrected $p$-values > 0.301). These results suggest that the more rewarding training policy was activated in the MRI signal from OTC about 5 s after trial onset and in the signal from DLPFC about 8.75 s after trial onset. The OTC signal also had transient information about the less rewarding past policy from about 5 s (Fig 3D, 3E).

To assess whether these effects were impacted by the choice made on each test trial, we re-ran the cluster tests above but excluded evidence from trials where the policy category matched the choice participants made. A significant decoding cluster for the more rewarding training policy was detected in OTC from 5 to 7.5 s (corrected $p = 0.003$) but no significant clusters were detected in DLPFC (candidate cluster at 5 s, corrected $p = 0.254$). The same results were seen for the less rewarding training policy. Significant decoding clusters were observed in OTC from 5 to 7.5 s (corrected $p = 0.003$) and from 11.25 to 12.5 s (corrected $p = 0.021$), but not in DLPFC (no candidate clusters). These results suggest that information about successful past policies could still be decoded from OTC when controlling for the choice made. This was not the case for DLPFC.

So far we have seen that OTC activated the optimal training policies and that DLPFC activated the more rewarding of the optimal training policies. These effects could be due to the neural reactivation of the policies from memory following the test cue, selective attention to specific policies when the response screen was shown, or both. To arbitrate between these possibilities, we examined decoding evidence locked to the response phase, and asked whether activation of the optimal training policies arose earlier than could be expected from the response phase alone (Fig 3F, 3G). The procedure for training the decoder remained the same as the previous analyses. However, the evaluation was conducted at each time point from 0 to 7.5 s relative to response phase onset rather than the cue onset. To establish the expected lag of policy decoding after the response phase onset, we examined decoding evidence on training trials using cluster-based permutation tests (window tested = 0–7.5 s, 10,000 permutations). Using these trials as a baseline revealed a significant decoding cluster for the optimal policy in OTC that was first detected 3.75 s after the start of the response phase during training trials (significant cluster window = 3.75–7.5 s, corrected $p < 0.001$, Fig 3F). In contrast to this expected lag, the cluster onset was shifted earlier on test trials, with significant information about the optimal training policies present in OTC already 0 s from the response phase onset (more rewarding training policy: significant cluster window = 0–7.5 s, corrected $p < 0.001$; less rewarding past policy: first significant cluster window = 0–2.5 s, corrected $p = 0.001$, second significant cluster window = 6.25–7.5 s, corrected $p = 0.014$). In line with these results, decoding evidence in OTC at 0 s from the response phase onset was significantly higher on test trials compared to training trials (more rewarding training policy: $M_{diff} = 2.94\%$,

SD$_{diff}$ = 3.38, $t$(37) = 5.29, corrected $p$ < 0.001; less rewarding training policy: $M_{diff}$ = 2.81%, SD$_{diff}$ = 3.97, $t$(37) = 4.31, corrected $p$ < 0.001). No differences at 0 s were detected for DLPFC (corrected $p$-values > 0.623, Fig 3G). To summarize, information about the optimal training policies was present from response phase onset in OTC during test trials. Due to the hemodynamic delay, this implies some reactivation of the optimal training policies in OTC before the response screen was displayed. Attention to specific stimuli shown on the response screen would have then plausibly contributed to the decoding effects in OTC from around 2.5 to 3.75 s after its onset (Fig 3F).

## Policy activation in OTC was associated with test choices

The neural results presented above suggest that OTC and DLPFC represented information predicted under an SF&GPI algorithm. To examine whether neural coding in OTC and DLPFC had a functional connection to participant choices, we correlated average decoding strength for the more rewarding training policy (the orange dots in Fig 3B) with the proportion of test trials in which participants generalized the more rewarding training policy. This revealed a significant positive correlation between neural activation of the more rewarding training policy in OTC and the implementation of that policy at test (Spearman's $Rho$ = 0.447, corrected $p$ = 0.010, Fig 3H). The equivalent correlation was not detected for DLPFC (Spearman's $Rho$ = 0.115, corrected $p$ = 0.492, Fig 3I). The correlation results held when using neural prioritization towards the more rewarding training policy (the orange dots in Fig 3C) as the neural variable (OTC: Spearman's $Rho$ = 0.433, corrected $p$ = 0.013; DLPFC: Spearman's $Rho$ = 0.161, corrected $p$ = 0.335). Neural coding in OTC was also directionally specific. Increased neural evidence for the more rewarding training policy in OTC showed a significant negative relation to the proportion of optimal MB choices at test (OTC: Spearman's $Rho$ = −0.375, corrected $p$ = 0.041; DLPFC: Spearman's $Rho$ = −0.098, corrected $p$ = 0.560). These results suggest that policy coding within the OTC was associated with test choices.

## Features could not be decoded on test tasks

Having seen that optimal training policies were activated on test tasks (prediction 1) and that the evidence for them was higher than the objective best policy (prediction 2), we tested whether features associated with the optimal policies were also represented (prediction 3). This required a different decoding approach for two reasons. The first was that feature triplets $\phi(1)$ and $\phi(4)$ were consistently associated with the optimal training policies and were thus selected on most trials. Training a feature decoder on these data directly would result in large imbalances in the number of trials per class. The second reason was that the feature triplets were correlated with the reward. Feature triplets $\phi(1)$ and $\phi(4)$ often resulted in a profit and feature triplets $\phi(2)$ and $\phi(3)$ often resulted in a loss.

To circumvent these issues, we trained feature decoders on fMRI data from a separate associative memory paradigm (see Session One in the Methods section and S10 Fig). Participants were first pre-trained on associations between visual cues and target stimuli. The target stimuli were feature numbers or cities that would later be used in the gem collector game. On each scanning trial, participants were shown a visual cue and needed to select its associated target from a choice array. Trials in which the correct number target was selected were used to train logistic decoders that could distinguish neural responses for the 12 number targets. The training process was similar to the process used for policy decoding. One TR per eligible trial was used as training input, taken 4–6 s after the response screen onset. The time shift and smoothing used for each ROI were the same as those used to decode policies. This approach could successfully recover the 12 number targets when tested in held-out data from the associative memory paradigm in all ROIs except MTL (S11 Fig). The trained decoders were then shown neural data from the test trials in the gem collector paradigm. This returned a cross-validated decoding probability for each feature number at each TR on each test trial. To ensure the decoding probabilities were stable, we repeated the procedure 100 times using random subsampling to match trial numbers ($M$ = 36 trials per training class, SD = 2.44, minimum = 29, maximum = 39). We then identified the three feature numbers associated with each of the optimal training policies and averaged the decoding probabilities for those number labels on each test trial.

Using this approach, we examined whether average decoding evidence for the three features anticipated under each of the optimal training policies was higher than chance (8.33% based on the 12 possible feature numbers). Like the earlier section on policies, decoding evidence was averaged from +2.5 s to +12.5 s following test trial onset. Feature information associated with the more rewarding training policy was not detected on test tasks (OTC: $M = 8.71\%$, SD = 1.12, $t(37)$ = 2.04, corrected $p = 0.392$; MTL: $M = 8.37\%$, SD = 0.40, $t(37)$ = 0.51, corrected $p > 0.99$; OFC: $M = 8.28\%$, SD = 0.52, $t(37)$ = −0.60, corrected $p > 0.99$; DLPFC: $M = 8.34\%$, SD = 0.72, $t(37)$ = 0.05, corrected $p > 0.99$). Equivalent results were found for features associated with the less rewarding training policy (OTC: $M = 8.61\%$, SD = 0.91, $t(37)$ = 1.87, corrected $p = 0.486$; MTL: $M = 8.38\%$, SD = 0.59, $t(37)$ = 0.47, corrected $p > 0.99$; OFC: $M = 8.37\%$, SD = 0.52, $t(37)$ = 0.48, corrected $p > 0.99$; DLPFC: $M = 8.38\%$, SD = 0.57, $t(37)$ = 0.50, corrected $p > 0.99$, S12 Fig).

## Discussion

This study aimed to investigate whether an SF&GPI algorithm could account for neural activity when humans transfer their experience from known to novel tasks. Behavioral results showed that human choices on new tasks relied on reusing policies that were successful in previous tasks. While this strategy was less optimal than a model-based process using a full model of the environment, past policies were applied in a reward-sensitive manner that led to high performance. These results were not due to a simple perseveration strategy (S5 Fig). An analysis of neural activity during test tasks showed that successful training policies were also represented in occipital-temporal and dorsolateral-prefrontal areas. These policies were prioritized as candidates for decision-making, with stronger activation than alternative policies that offered higher rewards. Activation strength in OTC was correlated with reuse behavior. We found no evidence for the reactivation of features associated with successful training policies. Our results speak towards a role of OTC and DLPFC in implementing an efficient form of generalization that is superior to model-free behavior, but less optimal than full model-based computation operating on a complete model of the environment.

### Computational accounts of generalization behavior

Consistent with previous behavioral research [7], a computational process based on SF&GPI could explain human generalization performance. However, it did not capture participant choices perfectly. This was evident in data showing that participants made fewer choices leading to feature triplet $\phi(4)$ on test trials than the model predicted (Fig 2G, 2H). Exploratory tests revealed that this was due to the presence of two distinct subgroups (S2 Fig). Half of the participants showed a full recapitulation of the SF&GPI predictions on test tasks. The other half showed a partial recapitulation. This suggests that some participants used different strategies on specific test tasks. When examining individual test tasks (S3 Fig), we further observed that the SF&GPI algorithm predicted the dominant choice in most cases. However, there was one test task on which choices were evenly split between the SF&GPI and MB predictions. Whereas most test tasks contained an anti-correlated structure in the feature weights that was similar to one or more training tasks, this outlier test task did not. This raises the possibility that at least some participants exploited structural similarities between the training and test tasks to determine their choices, similar to a Universal Value Function Approximator (UVFA) process in which similar task cues lead to similar rewards for a given action [28]. Tomov and colleagues observed evidence for UVFAs in a minority of subjects [7], but also had less structural overlap between training cues and the test cue. Future research should systematically manipulate the similarity between training and test scenarios to understand how this affects the computational process used for generalization.

Post hoc behavioral predictions of a UVFA process are included in S4 Fig. These predictions show consistencies with human choice, suggesting that UVFA could offer an alternative account of behavior. However, the neural findings from this experiment appear to be inconsistent with several UVFA predictions. Vanilla UVFAs predict equal activation of all available actions to compute and compare $Q$-values. In contrast, OTC showed activation of the optimal training actions/policies at choice onset, followed by sustained activation of the more rewarding action/policy (Fig 3D), whereas DLPFC

showed activation of the more rewarding action/policy (Fig 3E). While this could be explained by a partial Markov decision process, in which people only evaluate actions that were reinforced during training, it would be inconsistent with the UVFA-like empirical choice profile on test task $w$ = [1,1,−1] (S4B Fig), in which a non-reinforced action is selected with high frequency. A UVFA explanation of the empirical choice profile on test task $w$ = [1,1,−1] (S4B Fig) would also predict approximately equal neural activation of both the objective best policy and the more rewarding training policy. Instead, we find that the more rewarding training policy is activated significantly more than the objective best policy, for which decoding evidence is at chance (S9B Fig). Despite these neural inconsistencies, the behavioral similarities to UVFAs suggest that humans may utilize some kind of UVFA-like generalization, perhaps in addition to SF&GPI-like policy reuse, in which the evaluated actions are not explicitly represented in the brain regions considered here. Exploring potential hybrid strategies that combine UVFAs with SF&GPI could be a promising area of future investigation.

Recent studies have also shown behavioral and neural evidence for reinforcement learning (RL) based on configurations of cues, over and above their individual elements [29,30]. Applied to our setup, this would correspond to learning unique values for each (city, task) configuration. This account would not immediately capture the generalization effects reported here. While an agent learning to respond to each configuration would be expected to learn optimal solutions to the training tasks, it would not have an obvious mechanism to map this knowledge to new tasks (i.e., new configurations of cues), the same issue facing a model free-agent (S5 Fig). Nonetheless, configural accounts could in principle be extended to support generalization to novel configurations using function approximation. This would allow an agent to smoothly interpolate between the values for different (city, task) configurations experienced during training, to compute approximate values for the new (city, task) configurations on test trials. Interestingly, this is precisely what UVFAs do. While our experiment was designed to disambiguate between SF&GPI and MB control using a complete model, we do see signatures of UVFA-like response patterns (S4B Fig), even though our neural results appear to be inconsistent with UVFA predictions (S9B Fig). Exploring UVFAs as a generalization of configural RL in the multi-task domain could be an interesting direction for future investigation.

A final account of generalization behavior could be a heuristic strategy, in which participants focus exclusively on the feature with the highest positive reward weight. Participants could then retrieve a policy from the training tasks that shared this top ranked feature. Such a heuristic would lead to SF&GPI-like behavior in three test cases. However, it would not predict neural activation of the less rewarding training policy, as there would be no need to consider other features or policies once the top-ranked gem had been identified. Therefore, this heuristic strategy would not explain the neural effects in OTC, where both the more and less rewarding training policies were activated on test tasks (Fig 3C, 3D).

## Neural predictions of SF&GPI

Consistent with SF&GPI predictions, we observed neural prioritization of the optimal training policies during test tasks. This was most prominent in OTC. Evidence from our data suggests that two distinct sources contributed to decoding in this region. One was a reactivation of optimal training policies after seeing the test cue, as optimal training policies could be decoded earlier than what would be expected based on training trials. This could reflect expectations about upcoming stimuli in visual cortex [31,32], with added modulation based on behavioral relevance. The other source was prioritized processing of the optimal training policies shown on the response screen. This enhanced processing could reflect value-driven attentional capture, in which stimuli previously associated with reward are preferentially attended to in new contexts [33]. While OTC is not a usual candidate for experiments on RL and decision-making, we note recent reports that neural replay decoded from OTC relates to the formation of successor representations [25], which are one of the theoretical foundations of SF&GPI. Neural prioritization was also observed DLPFC, an area proposed to encode policies [11,12]. This finding aligns with DLPFC's role in context-dependent behavior [13–17], as the task cue on each trial can be seen as a context cue that determines the current response mapping. One interpretation of our findings is that DLPFC

can generalize this role outside a set of training cues, retrieving relevant response mappings for novel context cues with similar but non-identical structure to earlier contexts. We also note that although the prioritization seen in our data is consistent with an SF&GPI-like process, it could also support hybrid models that use cached option values to identify useful candidates for particular decisions, and then perform model-based planning on that subset of options [34,35].

Based on proposals that the MTL and OFC serve as predictive maps that encode information about future states [18–22], we predicted that features expected under the optimal training policies would be detected in these regions on test trials. We did not find evidence that this was the case. One possibility is that the features were not central to participants' decision process, which could occur if choices were primarily based on structural similarities between the training and test cues. It is also possible that the features were used but that we were unable detect them. This could occur if feature numbers were represented with different neural patterns in the localizer task used to train the neural decoders than the main task. Cross-validated tests with policy decoders imply some support for this possibility, suggesting that decoders trained and tested on the main task tended to perform better, compared to those trained on the localizer and then tested on the main task (S13 Fig). Future experiments should therefore develop designs in which feature information can be remapped between the optimal and suboptimal policies within each block of a single paradigm to train more sensitive feature decoders. A second reason for not detecting features could be because each test task was repeated five times per block without feedback. This could have meant that feature triplets were used to compute expected rewards during initial test trials but that a policy was cached and used for remaining test trial repeats. Decoding predictive feature representations remains a critical target in future neural tests of SF&GPI.

It is also possible that OFC and MTL were engaged during the experiment but encoded other variables. For instance, one influential view of OFC function is that the OFC represents the prospect of reward for different options during decision-making [36,37]. The present experiment was not optimized to decode value signals due to the wide variation and largely unsystematic distribution of the rewards across training and test tasks (Fig 4), a compromise that arose from the other factors we needed to balance in the experiment to test our central predictions. Some studies also suggest that hippocampal signaling may modulate the reinstatement strength of neural patterns in visual areas [38,39]. Based on these factors, one speculative possibility could be that value information in OFC and hippocampal signaling modulate OTC and DLPFC activity on new tasks, in favor of successful past solutions.

## Limitations

The present study had two main limitations. One was that each trial involved a single decision step. This meant that multi-step successor features were equivalent to single-step state features in the present design. This simplification was made to meet the practical constraints of fMRI scanning but future studies will need to devise practical ways to retain a multi-step element that can distinguish between computational processes using successor features and state features for generalization. This would also help to distinguish reuse based on SF&GPI from more subtle forms of MB control that operate on a partial model of the environment. The most direct data we have on whether participants had a partial model in the current experiment were feature estimates collected after fMRI scanning. Specifically, participants were asked to estimate the features (gem numbers) for each city in the final experimental block (S6 Fig). If participants had a partial model of the environment, one prediction is that participants should have more accurate knowledge about features associated with the optimal training policies. We did not detect robust evidence for this prediction (corrected $p$-values > 0.09, S6A Fig). However, we did observe a significant correlation between having less accurate knowledge about features associated with the suboptimal training policies and the reuse of successful past solutions during test tasks (Spearman's $Rho = 0.493$, $p = 0.016$, S6B Fig). One possibility based on this result is that participants with higher policy reuse had a partial model of the environment or a noisier memory for certain features. Unfortunately, the present design can only arbitrate between a computational process based on SF&GPI and a MB process that uses a complete model of the environment. It is unable to arbitrate between SF&GPI and forms of MB processing that use a partial model of the environment or have noisy memory. Multi-step designs

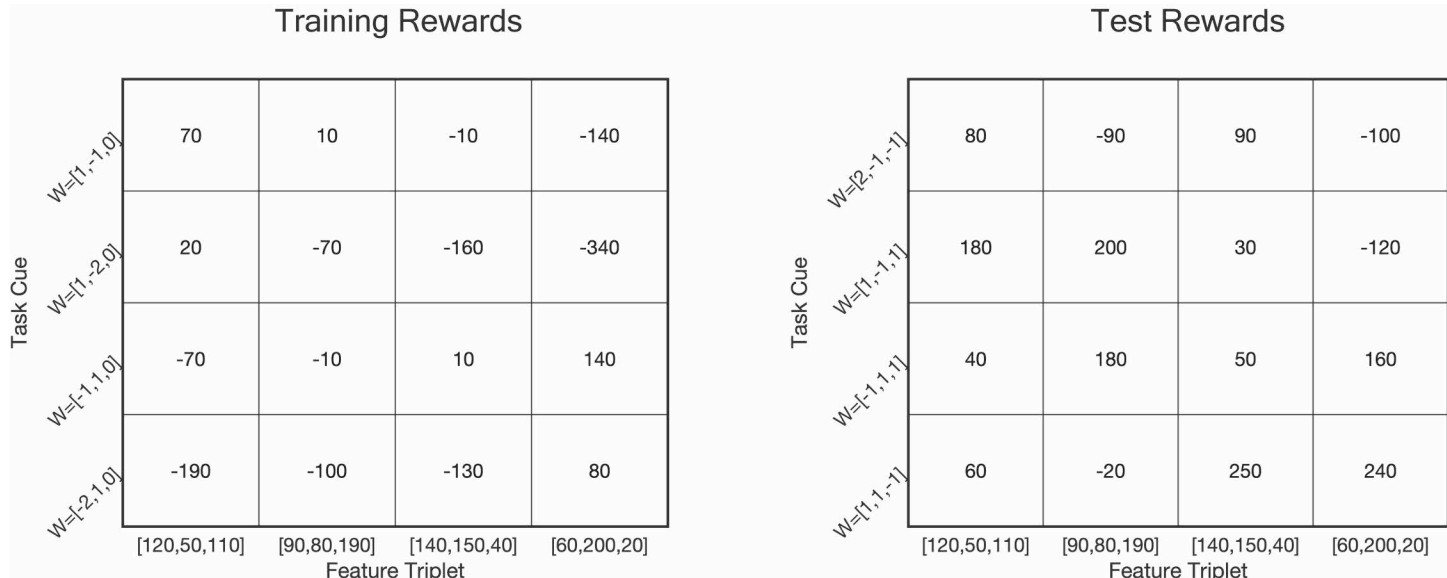

**Fig 4. Reward outcomes for each combination of task cue and policy in the experiment.** Rewards for training tasks are shown in the left-hand matrix. Rewards for test tasks are shown in the right-hand matrix. The order of feature triplets here is the same as other figures in the main text and supplement. [120,50,110] (or a permutation of these numbers) corresponds to 'Feature Triplet 1', [90,80,190] (or a permutation of these numbers) corresponds to 'Feature Triplet 2' and so on. Materials to reproduce this figure are available at https://gin.g-node.org/sam.hall-mcmaster/sfgpi-behavioural-analysis.

could better address the separation between SF&GPI and partial MB control in future experiments. An MB agent with a partial model should be able to report features for each individual state that comprises the optimal training policies. A memory-based SF&GPI agent is expected to fail on such queries, as it stores a compact summary of this information that does not have individual state resolution. A second limitation was that the reward for the objective best policy was only 10–20 points higher than the more rewarding training policy (from among the optimal training policies). While we observed a high degree of reuse as predicted under an SF&GPI algorithm, it could be that the computational process would differ—with more equal evaluation across all options—if the reward prospect for previously unsuccessful policies had been higher at test. This would align with evidence that more computationally intensive forms of decision-making, such as MB planning, are adopted when the reward for accurate performance is heightened [40]. The present experiment was not optimized to address meta-control between computational processes for different reward-effort trade-offs. Future research could therefore manipulate the difference in reward between successful and unsuccessful training policies during test tasks, to better understand the conditions under which an SF&GPI-like transfer process is (or is not) used.

## Conclusions

Overall, the present study provides behavioral and neural evidence that generalization to new tasks was more consistent with an SF&GPI-based algorithm than an MB algorithm using a full model of the environment. These results do not rule out an MB algorithm operating on a partial model of the environment, or with noisy memory for features associated with suboptimal training choices. SF&GPI and UVFAs should also be systematically compared in the presence and absence of structural similarity between training and test cues in future studies. Ultimately, this experiment found that successful past solutions were prioritized as candidates for decision-making on tasks outside the training distribution. This prioritization provides flexibility when faced with new decisions problems and has lower computational cost than considering all available options. These findings take a step towards illuminating the flexible yet efficient nature of human intelligence.

## Methods

### Ethics statement

The experimental protocol was approved by the ethics commission at the German Psychological Society (reference: SchuckNicolas2022-10-24AM) and conducted in accordance with the Declaration of Helsinki. Participants signed informed consent before each test session.

### Participants

Forty people participated in the experiment. One participant was excluded from the final sample due to low behavioral performance. The number of points earned in each scanning session was more than 2.5 standard deviations below the sample mean. Another participant was excluded due to excessive head motion. This was based on framewise displacement (FD), which measures the change in head position between adjacent data points [41]. All functional runs in the second scanning session for the participant were more than 2.5 standard deviations above the sample mean for FD. The resulting 38 participants were between 18 and 35 years of age (mean = 26 years, 23 female). All individuals had normal or corrected-to-normal vision and did not report an on-going neurological or psychiatric illness. €75 was paid for completing the experiment and €25 extra could be earned as a performance dependent bonus (€10 in session 1 and €15 in session two).

### Materials

Psychopy3 (RRID: SCR_006571, version 2021.2.3, [42]) and Pavlovia (RRID:SCR_023320, https://pavlovia.org/) were used to prescreen prospective participants. Stimulus presentation during the scan sessions was controlled using Psychophysics Toolbox-3 (RRID:SCR_002881, version 3.0.17) in MATLAB (RRID:SCR_001622). The sessions used stimuli from [43] and city images from Rijan Hamidovic, Aleksander Pesaric, Pixabay and Burst that were retrieved from https://pexels.com. Scan preparation was completed on a computer outside the scanner with a spatial resolution of 2048 × 1,152 and refresh rate of 60 Hz (MATLAB version R2021b). The main experimental tasks were run on a stimulus PC with a spatial resolution of 1,024 × 768 and a refresh rate of 60 Hz, which projected to a magnetically safe screen for participants inside the scanner (MATLAB version R2017b). Responses were recorded using two fiber optic button boxes (Current Designs, Philadelphia, PA, USA). Participants used two buttons on the left box (middle finger and index finger) and three buttons on the right box (index finger, middle finger and ring finger). Output files from the scanner were converted to the BIDs naming convention (RRID:SCR_016124) using ReproIn (RRID:SCR_017184, HeuDiConv version 0.9.0). Preprocessing and quality control were conducted using fMRIprep (RRID:SCR_016216, version 20.2.4, [44]) and MRIQC (RRID:SCR_022942, version 0.16.1, [45]). Behavioral analyses were performed in MATLAB (version 2021b). Neural analyses were performed in Python (RRID:SCR_008394, version 3.8.13), primarily using SciPy (RRID:SCR_008058, version 1.7.3), Pandas (RRID:SCR_018214, version 1.4.4), NumPy (RRID:SCR_008633, version 1.21.5), Matplotlib (RRID:SCR_008624, version 3.5.2), MNE (RRID:SCR_005972, version 1.4.2), Sklearn (RRID:SCR_019053, version 1.3.0), and Nilearn (RRID:SCR_001362, version 0.10.1). Data and code management was performed using DataLad (RRID:SCR_003931, version 0.17.9) and GIN (RRID:SCR_015864).

### Data and code

Data and code for this project are openly available in the following repositories (Table 1).

### Online prescreening

Participants completed an online prescreening task prior to study admission. This was conceptually similar to the main task in MRI session two. However, it had different stimuli, different state features, and a different theme to avoid biasing

participants during scanning. The market values used for the prescreening were: $w_{train}$ = {[1, −1, 0], [−1, 1, 0], [1, −2, 0], [−2, 1, 0]}. The state features were: $\phi$ = {[100, 0, 0], [30, 30, 160], [100, 100, 0], [0, 100, 70]}. The vector elements in each triplet were shuffled with a randomly selected permutation rule for each participant. The theme used for prescreening was from [7]. Participants completed 80 trials with equal numbers of each training task ($w_{train}$) in a random order. No test tasks ($w_{test}$) were presented. Trials had the same structure as Fig 1A but with different event timing. Market values were shown for 2.5 s, no response limit was imposed, and feedback was shown for 3 s. The cue-target interval was set to 0 s and the inter-trial interval (ITI) to 1 s. Prospective participants needed to achieve an average reward above zero points to pass the prescreening and participate in the experiment. Up to three attempts could be made. Ninety-five percent of people who passed the prescreening did so on their first attempt. Four prospective participants did not pass and were not admitted to the experiment.

## Session one

During the first experimental session, participants completed an associative retrieval task inside the MRI scanner. Session one was designed to produce a localizer dataset that could be used as training data for neural decoders. Data from this session were used to train feature decoders but not policy decoders in the present paper. For the procedure behind the gem collector game, see session two.

**Scan preparation.** In preparation for the first MRI scan, participants learned associations between cartoon objects (e.g., a balloon) and cities or number stimuli that would appear during session two. There were 4 cities (Sydney, Tokyo, New York, London) and 12 numbers (20, 40, 50, 60, 80, 90, 110, 120, 140, 150, 190, 200) in total. Each target was associated with two cartoon images, resulting in 32 associations. The associations between stimuli were randomly determined for each person. Participants completed a structured training procedure to learn the associations. The training began with the eight object-city associations. On each trial, participants were shown a cartoon object (e.g., a balloon) for 1.5 s, followed by a 0.25 s blank delay. The four cities were then shown on screen in a random order. Participants needed to press the key (D, F, J, K, or L) corresponding to the screen position of the correct city within 5 s. Following a response, feedback was shown for 2 s ('Correct!' or 'Incorrect!'). If the response was incorrect, participants were shown

**Table 1. Data and code resources.**

| Resource | Description | Link |
|---|---|---|
| sfgpi-task [46] | Behavioral tasks | https://gin.g-node.org/sam.hall-mcmaster/sfgpi-task |
| sfgpi-behavioral-data [47] | Behavioral data | https://gin.g-node.org/sam.hall-mcmaster/sfgpi-behavioural-data |
| sfgpi-behavioral-analysis [48] | Behavioral analyses | https://gin.g-node.org/sam.hall-mcmaster/sfgpi-behavioural-analysis |
| sfgpi-bids [49] | MRI data (BIDs format) | https://gin.g-node.org/sam.hall-mcmaster/sfgpi-bids |
| sfgpi-fmriprep [50] | MRI data (preprocessed) | https://gin.g-node.org/sam.hall-mcmaster/sfgpi-fmriprep |
| sfgpi-mriqc [51] | MRI data quality reports | https://gin.g-node.org/sam.hall-mcmaster/sfgpi-mriqc |
| sfgpi-masks [52] | Anatomical masks | https://gin.g-node.org/sam.hall-mcmaster/sfgpi-masks |
| sfgpi-neural-analysis [53] | Neural analyses | https://gin.g-node.org/sam.hall-mcmaster/sfgpi-neural-analysis |
| sfgpi-tools | Software tools | https://gin.g-node.org/sam.hall-mcmaster/sfgpi-tools |

the correct object-city pairing. The trial concluded with a blank ITI lasting 1 s. Each block had 8 trials in total. This included one trial per association (e.g., four cities with two associated objects each), presented in a random order. Participants continued to complete blocks until achieving 8/8 correct responses. To reduce training time, the object duration on screen was shortened from 1.5 to 0.5 s after the first block for each set of 8 associations. Using the same training structure, participants then learned 8 associations for each of the following number sets in turn: [20, 40, 50, 60], [80, 90, 110, 120], [140, 150, 190, 200]. To conclude the training, participants were tested on longer blocks with one trial for each of the 32 associations. Blocks continued until participants scored at least 90% correct (29/32 trials). After passing the 90% criterion once, participants had to do so one more time but with a shorter response deadline of 1.5 s.

**Scan task.** The MRI task for session one was a cued retrieval task. The setup was similar to the final training stage outlined in the previous section. On each trial, participants were shown a cartoon object for 0.5 s and asked to imagine its associated target (i.e., the city or number associated with it). Following a blank delay, participants were presented with four possible response options in a random order on screen and needed to select the associated item within 1.5 s. A feedback screen was then shown and the trial concluded with a blank ITI. The delay between the cartoon object offset and the response phase onset was drawn from a truncated exponential distribution, with a mean of 3 s (lower bound = 1.5 s, upper bound = 10 s). Feedback ('+10' or '+0') was shown for 0.5 s. The ITI was drawn from a truncated exponential distribution with a mean of 2 s (lower bound = 1.5 s, upper bound = 10 s). Participants completed 64 trials per block and 10 blocks in total. The trial order was constrained so that the correct target was not repeated on consecutive trials. Trials could also be labeled based on the following categories: cities (Sydney, Tokyo, New York, London), number set 1 (20, 40, 50, 60), number set 2 (80, 90, 110, 120) and number set 3 (140, 150, 190, 200). Consecutive trials with the same category were allowed to occur a maximum of two times per block. To motivate good performance, participants were shown their average accuracy, median RT and their bonus earnings up to that point at the end of each block. Fieldmaps and an T1w anatomical image were collected after the 5th block. This scan resulted in 640 retrieval trials, with 40 trials for each target stimulus.

## Session two

During the second experimental session, participants completed the gem collector game described in the main text. Sessions were held on consecutive days, with session two occurring the day after the session one.

**Scan preparation.** Participants completed a brief preparation task to orient them to the gem collector theme and trial events. This included 30 practice trials. Practice trials had the same event structure as those shown in Fig 1. The market values used in the preparation task were: $w_{train}$ = {[1, −1, 0], [−1, 1, 0]}. The state features were: $\phi$ = {[100, 0, 0], [30, 30, 160], [100, 100, 0], [0, 100, 70]}. The vector elements in each triplet were shuffled with a randomly selected permutation rule for each participant. The assignment between cities and state feature triplets was also randomized. Participants were told how the game profit was calculated on each trial and had to correctly calculate the profit for three example trials to confirm their understanding. The first 10 practice trials provided feedback after each choice. The remaining 20 trials consisted of 15 trials with feedback and 5 trials without feedback. Event timing was reduced across practice to prepare participants for the event speed during scanning. Market values were presented for 3.5 s, and a 5 s response deadline was imposed for the first 10 trials. This was reduced to 2 and 1.5 s for the remaining trials. The feedback (2.5 s), cue-target interval (1 s) and inter-trial interval (2 s) had consistent durations. Participants did not need to reach a performance threshold on the practice trials to proceed to the scan task.

**Scan task.** The cognitive task used in the second scanning session is described in the main text. On each trial, participants were shown the selling prices of three gem stimuli (2 s). This was followed by a jittered blank delay. The specific duration on each trial was drawn from a truncated exponential distribution with a mean of 3.5 s, a lower bound of 3 s, and an upper bound of 10 s. Four city stimuli then appeared on screen in a random order. The random ordering prevented participants from preparing a motor response before seeing the choice stimuli. Participants had up 1.5 s to

choose a city. Upon entering a response, a feedback screen was presented (2.5 s) that showed the selling prices, the number of gems in the selected city and the profit (or loss). The feedback screen was omitted on certain trials (described below). In the event that no response was made within 1.5 s, 'Too slow!' was printed to the screen as feedback. The trial concluded with a jittered ITI. The specific duration on each trial was drawn from a truncated exponential distribution with a mean of 3 s, a lower bound of 2.5 s, and an upper bound of 10 s.

Each event in the trial sequence had a specific function. The selling prices were used to define the 'task' on each trial. A task is a context that specifies which features of the environment should be prioritized during decision-making. For example, if square gems have a high selling price and the other gems sell for $0 per unit, the task on that trial is to choose the option with the highest number of square gems. In this way, the selling prices (which were numeric weight vectors) allowed us to manipulate the task on each trial in a precise manner. Another important trial event was the feedback. Each city was had unique number of gems. For example, Sydney might have 120 square gems, 50 triangle gems and 110 circle gems. The number of gems were state features ($\phi$) associated with each city. The following set was used during scanning: $\phi$ = {[120, 50, 110], [90, 80, 190], [140, 150, 40], [60, 200, 20]}. The profit (or loss) on each trial was the dot product of the task weight triplet (the selling prices) and the state feature triplet (the gem numbers in the selected city).

The experiment had two main trial types. Training trials allowed participants to learn about the state features for each city because they showed feedback after each choice. Test trials were used to assess participants' generalization strategy and did not provide feedback. Participants were informed that the reward from all trials counted towards the €15 performance bonus in the session, even if feedback was not shown. Training and test trials were also distinguished based on the tasks (selling prices) shown to participants. The training tasks used in the experiment were: $w_{train}$ = {[1, −1, 0], [−1, 1, 0], [1, −2, 0], [−2, 1, 0]}. The test tasks were: $w_{test}$ = {[2, −1, −1], [−1, 1], [1, −1, 1], [1, 1, −1]}. The reward for enacting each policy in each of the tasks is shown in Fig 4.

Each block had 68 trials. The order of training and test trials were pseudo-randomized based on a set of constraints. The block was divided into two phases. The first 32 trials were training trials (phase 1). The remaining 36 trials were a mixture of 16 training trials and 20 test trials (phase 2). Trials in the first phase were constrained so that: (1) equal numbers of the four training tasks were shown, (2) the same task was not repeated on consecutive trials and (3) trials with a particular optimal choice under SF&GPI (e.g., Sydney) were preceded by an equal number of trials with the same or a different optimal choice (e.g., 50% Sydney and 50% New York). Equivalent constraints were applied to the remaining trials (i.e., phase 2). Trials in phase 2 had the additional constraints that: (4) the phase did not begin with a test trial and (5) no more than 3 test trials were presented in a row. The two phases in each block were recorded as separate fMRI runs to aid leave-one-run-out cross-validation (described in the section on neural analyses).

Participants completed 6 blocks in total (12 fMRI runs). There were six possible permutation rules that could be used to order the numeric elements within each task and state feature triplet: {[1, 2, 3], [1, 3, 2], [2, 1, 3], [2, 3, 1], [3, 1, 2], [3, 2, 1]}. One configuration was used for each block. The configuration sequence across blocks was randomized. The state features for each city were also varied across blocks. We ensured that each city was paired with each state feature triplet at least once. A random mapping was used for the remaining two blocks. The mapping was pseudo-randomized across blocks so that no city had the same state feature triplet in consecutive blocks. The screen position of the three gem stimuli was randomized for each participant but remained consistent throughout the experiment. Permuting the triplet elements and changing the state features in each city across blocks preserved the logical structure of the experiment, while giving the appearance of new game rounds for participants.

**Post scan.** To conclude session two, participants were asked to estimate the number of triangle, square, and circle gems that had appeared in each city during the final block of the scan task. Estimates were restricted to a range between 0 and 250 gems. Participants were also asked to rate their confidence for each estimate on a scale from 0 (not confident at all) to 100 (completely confident). These measures were not probed during the scan itself to avoid biasing participants' learning strategy.

## MRI parameters

MRI acquisition parameters were based on [24]. Data were acquired using a 32-channel head coil on a 3-Tesla Siemens Magnetom TrioTim MRI scanner. Functional measurements were whole brain T2* weighted echo-planar imaging (EPI) with multi-band acceleration (voxel size = 2 × 2 × 2 × mm; TR = 1250 ms; echo time (TE)=26 ms; flip angle (FA)=71°; 64 slices; matrix = 96 × 96; FOV = 192 × 192 mm; anterior-posterior phase encoding direction; distance factor = 0%; multi-band acceleration = 4). Slices were tilted +15 relative to the corpus callosum, with the aim of balancing signal quality from MTL and OFC [54]. Functional measurements began after five TRs (6.25 s) to allow the scanner to reach equilibrium and help avoid partial saturation effects. Up to 401 volumes (session one) or 333 volumes (session two) were acquired during each task run; acquisition was ended earlier if participants had completed all trials. Fieldmaps were measured using the same scan parameters, except that two short runs of 20 volumes were collected with opposite phase encoding directions. Fieldmaps were later used for distortion correction in fMRIPrep (RRID: SCR_016216, version 23.0.2, [44]). Anatomical measurements were acquired using T1 weighted Magnetization Prepared Rapid Gradient Echo (MPRAGE) sequences (voxel size = 1 × 1 ×1 × mm; TR = 1900 ms; TE = 2.52 ms; FA = 9°; inversion time (TI)=900 ms; 256 slices; matrix = 192 × 256; FOV = 192 × 256 mm).

## MRI preprocessing

Results included in this manuscript come from preprocessing performed using *fMRIPrep* 23.0.2 (RRID:SCR_016216, [44,55]), which is based on *Nipype* 1.8.6 (RRID:SCR_002502, [56,57]).

**Preprocessing of $B_0$ inhomogeneity mappings.** A total of 2 fieldmaps were found available within the input BIDS structure per subject. A $B_0$-nonuniformity map (or *fieldmap*) was estimated based on two (or more) EPI references with topup (FSL 6.0.5.1:57b01774; [58]).

**Anatomical data preprocessing.** A total of 2 T1-weighted (T1w) images were found within the input BIDS dataset per subject. All of them were corrected for intensity non-uniformity (INU) with N4BiasFieldCorrection [59], distributed with ANTs 2.3.3 (RRID:SCR_004757, [60]). The T1w-reference was then skull-stripped with a *Nipype* implementation of the antsBrainExtraction.sh workflow (from ANTs), using OASIS30ANTs as target template. Brain tissue segmentation of cerebrospinal fluid (CSF), white-matter (WM), and gray-matter (GM) was performed on the brain-extracted T1w using fast (FSL 6.0.5.1:57b01774, RRID:SCR_002823, [61]). An anatomical T1w-reference map was computed after registration of 2 T1w images (after INU-correction) using mri_robust_template (FreeSurfer 7.3.2, [62]). Brain surfaces were reconstructed using recon-all (FreeSurfer 7.3.2, RRID:SCR_001847, [63]), and the brain mask estimated previously was refined with a custom variation of the method to reconcile ANTs-derived and FreeSurfer-derived segmentations of the cortical gray-matter of Mindboggle (RRID:SCR_002438, [64]). Volume-based spatial normalization to two standard spaces (MNI152NLin6Asym, MNI152NLin2009cAsym) was performed through nonlinear registration with antsRegistration (ANTs 2.3.3), using brain-extracted versions of both T1w reference and the T1w template. The following templates were selected for spatial normalization and accessed with *TemplateFlow* (23.0.0, [65]): *FSL's MNI ICBM 152 non-linear 6th Generation Asymmetric Average Brain Stereotaxic Registration Model* (RRID:SCR_002823; TemplateFlow ID: MNI152NLin6Asym, [66]), *ICBM 152 Nonlinear Asymmetrical template version 2009c* (RRID:SCR_008796; TemplateFlow ID: MNI152NLin2009cAsym, [67]).

**Functional data preprocessing.** For each of the 22 BOLD runs found per subject (across all tasks and sessions), the following preprocessing was performed. First, a reference volume and its skull-stripped version were generated using a custom methodology of *fMRIPrep*. Head-motion parameters with respect to the BOLD reference (transformation matrices, and six corresponding rotation and translation parameters) are estimated before any spatiotemporal filtering using mcflirt (FSL 6.0.5.1:57b01774, [41]). The estimated *fieldmap* was then aligned with rigid registration to the target EPI (echo-planar imaging) reference run. The field coefficients were mapped on to the reference EPI using the transform. BOLD runs

were slice-time corrected to 0.588 s (0.5 of slice acquisition range 0–1.18 s) using 3dTshift from AFNI (RRID:SCR_005927, [68]). The BOLD reference was then co-registered to the T1w reference using bbregister (FreeSurfer) which implements boundary-based registration [69]. Co-registration was configured with six degrees of freedom. Several confounding time-series were calculated based on the *preprocessed BOLD*: FD, the derivative of the root mean squared variance over voxels (DVARS), and three region-wise global signals. FD was computed using two formulations following Power (absolute sum of relative motions, [70]) and Jenkinson (relative root mean square displacement between affines, [41]). FD and DVARS are calculated for each functional run, both using their implementations in *Nipype* (following the definitions by [70]). The three global signals are extracted within the CSF, the WM, and the whole-brain masks. Additionally, a set of physiological regressors were extracted to allow for component-based noise correction (*CompCor*, [71]). Principal components are estimated after high-pass filtering the *preprocessed BOLD* time-series (using a discrete cosine filter with 128 s cutoff) for the two *CompCor* variants: temporal (tCompCor) and anatomical (aCompCor). tCompCor components are then calculated from the top 2% variable voxels within the brain mask. For aCompCor, three probabilistic masks (CSF, WM, and combined CSF + WM) are generated in anatomical space. The implementation differs from that of Behzadi and colleagues in that instead of eroding the masks by 2 pixels on BOLD space, a mask of pixels that likely contain a volume fraction of GM is subtracted from the aCompCor masks. This mask is obtained by dilating a GM mask extracted from the FreeSurfer's *aseg* segmentation, and it ensures components are not extracted from voxels containing a minimal fraction of GM. Finally, these masks are resampled into BOLD space and binarized by thresholding at 0.99 (as in the original implementation). Components are also calculated separately within the WM and CSF masks. For each CompCor decomposition, the *k* components with the largest singular values are retained, such that the retained components' time series are sufficient to explain 50 percent of variance across the nuisance mask (CSF, WM, combined, or temporal). The remaining components are dropped from consideration. The head-motion estimates calculated in the correction step were also placed within the corresponding confounds file. The confound time series derived from head motion estimates and global signals were expanded with the inclusion of temporal derivatives and quadratic terms for each [72]. Frames that exceeded a threshold of 0.5 mm FD or 1.5 standardized DVARS were annotated as motion outliers. Additional nuisance timeseries are calculated by means of principal components analysis of the signal found within a thin band (*crown*) of voxels around the edge of the brain, as proposed by [73]. The BOLD time-series were resampled into standard space, generating a *preprocessed BOLD run in MNI152NLin6Asym space*. First, a reference volume and its skull-stripped version were generated using a custom methodology of *fMRIPrep*. The BOLD time-series were resampled onto the following surfaces (FreeSurfer reconstruction nomenclature): *fsnative*. All resamplings can be performed with *a single interpolation step* by composing all the pertinent transformations (i.e., head-motion transform matrices, susceptibility distortion correction when available, and co-registrations to anatomical and output spaces). Gridded (volumetric) resamplings were performed using antsApplyTransforms (ANTs), configured with Lanczos interpolation to minimize the smoothing effects of other kernels [74]. Non-gridded (surface) resamplings were performed using mri_vol2surf (FreeSurfer).

Many internal operations of *fMRIPrep* use *Nilearn* 0.9.1 (RRID:SCR_001362, [75]), mostly within the functional processing workflow. For more details of the pipeline, see the section corresponding to workflows in *fMRIPrep*'s documentation.

## Computational models

Theoretical predictions for choices on test tasks were generated using code from [7]. The code was adapted to the current task design, which included changing the training tasks, test tasks, state features, and the state space. Model training procedures remained the same.

The models were situated in a standard RL formalism called a Markov Decision Process $M = (S, A, p, r, \gamma)$, where $S$ is a set of states, $A$ is a set of actions, $p(s' | s, a)$ is the probability of transitioning to a subsequent state ($s'$) from an initial state ($s$) after taking action $a$, $r(s)$ is the reward received when entering state $s$, and $\gamma$ is a discount factor between 0 and 1 that down-weights future rewards. The environment also contained state features $\phi(s)$ and tasks $w$. Policies defining the action an agent takes in each state are denoted $\pi$.

**Model-based (MB).** The MB algorithm had perfect knowledge about the environment. To decide what to do on each test task, it computed expected values for all possible actions. Each expected value was defined as:

$$Q_w(s, a) = \sum_{s'} p(s' \mid s, a)\, r(s)$$

The experimental design used deterministic transitions and thus each entry in $p(s' \mid s, a)$ was 0 or 1. The reward, $r(s)$, was computed as:

$$r(s) = \phi(s)^T w$$

The MB algorithm selected the policy on the current test task with the highest expected value:

$$\pi_w = argmax_a Q_w(s, a)$$

**Successor features and generalized policy improvement (SF&GPI).** The SF&GPI algorithm is designed to learn estimates of the state features called successor features:

$$\psi^\pi(s, a) = \phi(s) + \sum_{s'} p(s' \mid s, a)\gamma\phi(s')$$

Since no further action could be taken in the terminal state (s'), this simplified to:

$$\psi^\pi(s, a) = \sum_{s'} p(s' \mid s, a)\phi(s')$$

This simplification meant that successor features were equivalent to state features in the present design. The SF&GPI algorithm then performs generalized policy improvement, which computes a policy for the current test task based on earlier training policies. The first step in this process is to compute expected values on the current test task under a set of earlier policies $\{\pi_1 \ldots \pi_n\}$:

$$Q_w^{\pi_i}(s, a) = \psi^{\pi_i}(s, a)^T w$$

An interesting property of the SF&GPI algorithm is that it is flexible with respect to the policies it considers from training. When considering all policies experienced during training, an SF&GPI algorithm makes the same predictions as an MB algorithm in the present design. However, when generalized policy improvement is restricted to the optimal training policies, its predictions on test tasks differ:

$$\pi_w = argmax_{\pi \in \Pi} Q_w^\pi(s, a)$$

Here, the set of policies, $\Pi$, contains the optimal training policies but not all policies. We used this formulation of SF&GPI to distinguish between a model of generalization that evaluates optimal training solutions on test tasks and an MB generalization process that evaluates all possible solutions.

### Behavioral analyses

Seventeen statistical tests are reported in the main text for behavioral data. The corresponding *p*-values reported were therefore corrected for 17 tests using the Bonferroni-Holm correction [76]. Auxiliary tests presented in the supplement but

mentioned in the main text were separately corrected using Bonferroni-Holm. Supplementary figure legends provide more details about the correction applied in these cases.

**Average performance.** The average points per choice on training trials was computed for each participant. To compare this performance against a sensible baseline, we simulated an equivalent sample of 38 agents that made random choices on the same trials. To ensure comparability, points from random choices were replaced with 0 points for trials where the equivalent human participant omitted a response. The average points per choice was calculated for each agent. The two distributions (human and agent) were then compared using an independent two-tailed $t$ test. This process was repeated to assess average performance on the test trials.

**State analysis.** The present experiment contained four terminal states. Each terminal state was defined by a unique feature triplet from the following set: $\phi$ = [120, 50, 110], [90, 80, 190], [140, 150, 40], [60, 200, 20]. The order of individual elements within each triplet varied across blocks in the experiment. The most rewarding terminal states during training trials were associated with $\phi(1)$ or $\phi(4)$. To assess whether participants made more choices leading $\phi(1)$ or $\phi(4)$, we recorded the proportion of choices leading to each terminal state across all training trials in the experiment. We then compared the proportions between each pair of states using paired two-tailed $t$ tests. The one exception was that we did not compare the choice numbers between $\phi(2)$ and $\phi(3)$, as this comparison was less related to the computational modeling predictions. To assess whether the same choice profile was recapitulated at test, the process above was repeated for test trials.

**Reuse behavior.** To further quantify the reuse of policies leading to $\phi(1)$ or $\phi(4)$ on test tasks, we computed the proportion of test trials in which choices to $\phi(1)$ or $\phi(4)$ were selected for each participant. To assess whether the proportions were above chance, the distribution was compared to a mean of 0.5 in a one-sample two-tailed $t$ test. To understand whether policy reuse was sensitive to prospective rewards, we computed the proportion of test trials in which participants selected the more rewarding optimal policy from training. While the previous analysis counted choices to either $\phi(1)$ or $\phi(4)$ regardless of outcome, this analysis counted choices to $\phi(1)$ or $\phi(4)$ only when the more rewarding of the two was selected. Since participants had a ¼ chance of selecting the more rewarding optimal training policy at random from the response array, the empirical proportions were compared to a mean of 0.25 in a one-sample two-tailed $t$ test. The same process was used to quantify how often the optimal policy was selected on training trials.

## Neural analyses

Decoding was performed separately for each ROI. The voxel activity on each trial was extracted from the current ROI and smoothed 2–4 mm. The specific smoothing was determined in validation analyses that did not use the trials needed to test our hypothesis. The resulting data were detrended, $z$-scored, and motion confounds (identified by fMRIPrep) were removed. Data cleaning was performed separately for each functional run using Nilearn signal clean. Trials without a response were then removed. The decoding procedure next entered a cross-validation loop. One block was held out to use as test data. The remaining blocks were used to train the decoder. A random subsample was taken to match the number of trials for each training class. It is important to note that only trials with feedback were used for decoder training (see Fig 1A). Voxel activity 4–6 s following feedback onset was extracted, and the city shown on the feedback screen was used as the training label. The specific time lag was determined in validation analyses (see S7 Fig). The resulting voxel patterns and labels were used to train a logistic regression model. This was implemented in Sklearn with an L2 penalty term and a one-vs-the-rest multiclass strategy. For more details, please see the Sklearn documentation. The trained decoder was then tested on each trial in the held-out run. Testing was performed at multiple TRs on each trial, from cue onset ($TR_0$) through to 15 s after it ($TR_{+12}$). This resulted in one probability score per city stimulus at each TR in the held-out test trial. Each score was the probability—according to the model—that a particular city was the correct label for the neural data provided. The class probabilities at a specific data point did not need to sum to 1 (due to the one-vs-the-rest approach). The cross-validation procedure was repeated until all functional runs had been used as held-out test data. For robustness,

the analysis procedure was repeated 100 times with random subsampling on each iteration. The results were based on model probabilities averaged across iterations. These probabilities are interpreted as decoding evidence throughout the paper.

**ROIs.** To extract voxel patterns from specific regions, we generated masks for the four ROIs using FreeSurfer cortical parcellations. Each mask was composed of one or more bilateral parcellations. Masks were specific to each participant. DLPFC was based on the rostral middle frontal gyrus parcellation. MTL combined the hippocampus, parahippocampal gyrus and entorhinal cortex parcellations. OFC combined the medial and lateral OFC parcellations. OTC combined the cuneus, lateral occipital, pericalcarine, superioparietal, lingual, inferioparietal, fusiform, inferiotemporal, parahippocampal and middle temporal parcellations. For more information, please see [77] and the FreeSurfer documentation: https://surfer.nmr.mgh.harvard.edu/fswiki/FsTutorial/AnatomicalROI/FreeSurferColorLUT.

**Validation.** To validate the decoding pipeline, we trained neural decoders to classify the city observed during feedback on training trials. We then tested how well we could recover the chosen city on held out training trials. One functional run was held out at a time to assess decoding performance. As training trials were not related to our hypothesis, we used the validation procedure to explore three possible time shifts ($+4\,s, +5\,s, +6\,s$ seconds following feedback onset) and three possible smoothing levels ($0\,mm, 2\,mm, 4\,mm$) that could be used to optimize decoder training. The validation results were based on 25 iterations. Data were subsampled at random on each iteration to match the number of trials from each class. The highest performing decoding parameters were selected for each ROI based on the validation results (S7 Fig). These parameters were later used when testing our hypothesis about neural reactivation during test tasks.

**Policy decoding.** The choice on each test trial could fall into one of four categories: (1) the more rewarding policy (among the optimal training policies); (2) the less rewarding policy (among the optimal training policies); (3) the objective best policy; and (4) the remaining policy. To examine the neural evidence for these categories, we extracted decoding probabilities for the four choice stimuli on test trials—that is, on trials that did not show feedback (Fig 1B). This extraction was first performed for the time point closest to cue onset ($TR_0$). The four decoding probabilities on each test trial at this time point—one for each choice stimulus—were divided into the four categories above. To avoid contamination from the previous trial, probabilities were removed in cases where the policy category on the current test trial was the same as the previous trial. Once this operation had been performed for all test trials, the probabilities in each category were averaged over trials. This process was repeated at each time point until 15 s after cue onset ($TR_{+12}$). The output of this process was one decoding time course per policy category, per participant. These were stored in a 38 × 4 × 12 matrix, in which dimensions were participants × categories × time points (TRs). A separate matrix was generated for each ROI. To test the average decoding strength on test trials against chance, we averaged the matrices above from 2.5 s after cue onset ($TR_{+2}$) through to 12.5 s ($TR_{+10}$). This time window was selected based on the average test trial duration of 10 s (Fig 1B) and the hemodynamic delay. The 2.5 s starting point was selected based on validation analysis (S7 Fig), which showed that an event often took two TRs to have a discernible impact on the decoding probabilities. The average probabilities for the optimal training policies were compared to the chance rate of 0.25, using one-sample two-tailed t-tests. The resulting *p*-values were corrected for 8 comparisons (2 policy categories × 4 ROIs) using the Bonferroni-Holm correction. To assess the evidence for the optimal training policies within OTC and DLPFC in a time-resolved manner, we conducted one-sample cluster-based permutation tests against a population mean of 0.25 (window tested = 2.5–12.5 s) using mne.stats.permutation_cluster_1samp_test. To assess neural prioritization in each ROI, we took the average decoding evidence for each of the optimal training policies and subtracted the average evidence for the objective best policy on test tasks. Prioritization scores were tested against for significance using one-sample two-tailed *t* tests. The two prioritization scores within each ROI were further compared with two-sample two-tailed *t* tests. The resulting p-values were corrected for 12 comparisons (2 prioritization scores × 4 ROIs and 4 follow-up tests) using the Bonferroni–Holm correction.

**Control analyses.** We performed two control analyses to examine the policy results from OTC and DLPFC. To distinguish between neural reactivation and processing of options during the response screen, we re-ran the decoding

analysis but tested the decoder on time points locked to the response phase onset. To be cautious about interpreting evidence at the onset itself, we set $TR_0$ to be the earlier TR closest to each response phase onset (rather than rounding to the closest TR which could potentially contain signal further into the response phase). We then performed cluster-based permutation tests from 0 s to +7.5 s relative to the response phase onset. Trials with feedback were used to establish a baseline time lag for the decoding of policy information. This was contrasted with the time course observed on trials without feedback. To control for the impact of choice on the core results, we examined decoding evidence for the optimal training policies on test trials where participants did not select the corresponding policy. For example, when extracting decoding probabilities for the more rewarding policy (among the optimal training policies), we extracted probabilities from trials where participants selected a different policy. We then repeated statistical tests for the average decoding evidence and cluster-based permutation tests across time. The Bonferroni-Holm correction was applied for the four statistical tests comparing average decoding strength against chance (2 policy categories × 2 ROIs).

**Feature decoding.** To decode feature information, logistic decoders were trained on fMRI data described in *Session One*. Trials in which participants selected the correct number target were eligible as training data. The response phase onset was taken for each eligible trial and time shifted 4–6 s. The closest TR to this time point was used the neural training data from that trial. The training label was the selected number target. Training data was locked to the response phase (rather than feedback events for policy decoding) because feedback was only presented on error trials in session one. The time shift and spatial smoothing used were the same as policy decoding. After training, the logistic decoders were evaluated on each TR from the test trials from *Session Two.* This resulted in a separate decoding probability for the 12 feature numbers at each TR on each test trial. We then iterated over the test trials and extracted the decoding probabilities for the features associated with more (or less) rewarding training policy and averaged them. Feature probabilities were excluded from the average in cases where a feature number was the same as one of the optimal rewards from training. Probabilities were also excluded in cases where the features being decoded were the same as those selected on the previous trial. These control measures were intended to get an estimate of feature evidence that was independent from the previous trial and could not be explained as a reward prediction. The remaining procedure was the same as policy decoding. The extracted probabilities were averaged over test trials for each ROI and then averaged from 2.5 s after cue onset through to 12.5 s. The average probabilities for features associated with the optimal training policies were compared to a chance rate of 1/12, using one-sample two-tailed t-tests. The resulting *p*-values were corrected for 8 comparisons (2 policy categories × 4 ROIs) using the Bonferroni–Holm correction.

**Associations with reuse behavior.** Spearman correlations were used to test whether policy decoding effects were associated with choices during the experiment. The average decoding evidence for the more rewarding training policy was correlated with the proportion of test trials on which participants made the choice predicted by SF&GPI. The decoding evidence had a precautionary measure to avoid contamination from the previous trial (see the earlier section on Policy Decoding). For consistency, the behavioral proportion used in correlation tests was calculated with the same trials used for the decoding evidence. The correlation was run twice, once for OTC and once for DLPFC. The process was then repeated but using neural prioritization as the neural measure (the difference in decoding evidence for the more rewarding training policy and the objective best policy). The original process was then repeated one more time, but using the proportion of model-based choices on test tasks as the behavioral measure. As correlation tests within each ROI did not use fully independent data, *p*-values were corrected for the two ROIs tested using the Bonferroni-Holm correction.

## Supporting information

**S1 Fig. Learning dynamics on training tasks.** This plot shows the proportion of optimal choices for the 48 training trials per block (i.e., as a function of time). The black line shows the mean across participants and shading indicates the standard error of mean. The red line indicates the point in each block when test tasks are introduced. The earliest this could occur was trial 34. Performance begins around chance ($M_{trial\ 1} = 0.24$, $SD_{trial\ 1} = 0.16$) and reaches an asymptote by the time

test tasks are introduced ($M_{\text{trial 33}} = 0.93$, $SD_{\text{trial 33}} = 0.10$). Materials to reproduce this figure are available at https://gin.g-node.org/sam.hall-mcmaster/sfgpi-behavioural-analysis.
(TIF)

**S2 Fig. Subgroup analysis.** Exploratory analysis of participant choices on test tasks revealed two subgroups. Each plot shows the proportion of choices on test tasks (y-axis) that lead to each feature triplet (x-axis). π* denotes feature triplets associated with optimal test policies. **A**: The predicted choice profile across test tasks according to an SF&GPI algorithm. **B**: The choice profile for a subgroup of participants ($n = 17$) showing a strong recapitulation of the SF&GPI predictions. The choice profile for the remaining participants ($n = 19$) showing a partial recapitulation. **B, C**: Dots show individual participant data. Lines connect individual participant data across the feature triplets. A–C: Materials to reproduce this figure are available at https://gin.g-node.org/sam.hall-mcmaster/sfgpi-behavioural-analysis.
(TIF)

**S3 Fig. Choice behavior on individual test tasks.** Participants' most common action matched the SF&GPI predictions in 3/4 cases. Each plot shows the proportion of choices (y-axis) that lead to each feature triplet (x-axis). Each panel shows the choice profile for an individual test task. The cue for an individual task was a weight vector, $w$, which is indicated in the plot titles. These cues were called market values during the gem collector game. Each panel includes three plots: the theoretical predictions of a model-based algorithm, the theoretical predictions of an SF&GPI algorithm, and empirical choices from human participants. Bars indicate the standard error of the mean in plots with human data. To formally examine human choice profiles on an individual task level, we compared whether participants made the SF&GPI predicted choice significantly more often than the alternatives. For example, using data from the test task $w = [2, −1, −1]$, we compared whether the proportion of choices leading to feature triplet 1 was significantly higher than the proportions leading to feature triplets 2, 3, and 4. Repeating this process for each individual test task revealed that participants made the SF&GPI-predicted choice significantly more often in all cases except one (all $t$-values without the exception > 4.54, all corrected $p$-values without the exception < 0.001). The exception can be seen in panel **B** where the proportion of SF&GPI-predicted choices and MB-predicted choices were comparable ($M_{\phi(4)} = 40.24\%$ of test trials, $SD_{\phi(4)} = 27.40$, $M_{\phi(3)} = 41.83\%$, $SD_{\phi(3)} = 32.91$, $t(37) = −0.166$, corrected $p = 0.869$). Choice proportions were compared using paired samples $t$-tests, and $p$-values were corrected for the 12 tests in this section using the Bonferroni–Holm correction. Materials to reproduce this figure are available at https://gin.g-node.org/sam.hall-mcmaster/sfgpi-behavioural-analysis.
(TIF)

**S4 Fig. Predictions from a Universal Value Function Approximator (UVFA) algorithm.** Each plot shows the proportion of choices (y-axis) that lead to each feature triplet (x-axis). Each panel shows the choice profile for an individual test task. The cue for an individual task was a weight vector, $w$, which is indicated in the plot titles. These cues were called market values during the gem collector game. Each panel includes three plots: the theoretical predictions of a UVFA algorithm, the theoretical predictions of an SF&GPI algorithm, and empirical choices from human participants. Bars indicate the standard error of the mean in plots with human data. UVFA predictions were generated post hoc using the implementation in [7]. Materials to reproduce this figure are available at https://gin.g-node.org/sam.hall-mcmaster/sfgpi-behavioural-analysis.
(TIF)

**S5 Fig. Choices on individual test tasks were not explained by model-free perseveration.** A model-free algorithm learns the optimal policies on training tasks but has no way to evaluate their success on test tasks. The result is that the algorithm reuses the optimal training policies in an unselective fashion. Each plot shows the proportion of choices (y-axis) that lead to each feature triplet (x-axis). Each panel shows the choice profile for an individual test task. The cue for an individual task was a weight vector, $w$, which is indicated in the plot titles. These cues were called market values during the gem collector game. Each panel includes three plots: the theoretical predictions of a model-free algorithm, the theoretical

predictions of an SF&GPI algorithm, and empirical choices from human participants. Bars indicate the standard error of the mean in plots with human data. Materials to reproduce this figure are available at https://gin.g-node.org/sam.hall-mcmaster/sfgpi-behavioural-analysis.
(TIF)

**S6 Fig. Post scan feature estimates and confidence ratings.** Participants were asked to estimate the number of triangle, square, and circle gems that had appeared in each city during the final block of the scanning task. Estimates were restricted to a range between 0 and 250 gems. Participants were also asked to rate their confidence for each estimate on a scale from 0 (not confident at all) to 100 (completely confident). **A**: The mean absolute error (y-axis) for each feature triplet (x-axis). The mean error in each bar is an average across the three estimates provided for each triplet. No significant differences in mean absolute error were detected when correcting for multiple comparisons (all corrected $p$-values > 0.09). **B**: The relationship between the mean error for feature triplets associated with the suboptimal training policies and policy reuse. Feature triplets 2 and 3 were associated with suboptimal training policies. Policy reuse is the proportion of test trials in which the more rewarding optimal training policy was selected. We detected a significant correlation between these variables after correcting multiple comparisons, suggesting that individuals with less accurate estimates of features linked to the suboptimal training policies were less likely to select those policies on test trials. **C**: The same plot as B, except showing the mean absolute error for feature triplets associated with the optimal training policies. **D**: Mean confidence ratings (y-axis) for each feature triplet (x-axis). Mean confidence in each bar is an average across the three ratings provided for each triplet. Participants were significantly more confident in their ratings for feature triplets associated with the optimal training policies [$\phi(1)$ and $\phi(4)$] compared to $\phi(3)$ which was associated with one of the suboptimal training policies ($t(37)_{\phi(1) \text{ vs. } \phi(3)}$ = 3.08, corrected $p$ = 0.046; $t(37)_{\phi(4) \text{ vs. } \phi(3)}$ = 3.91, corrected $p$ = 0.0054, all remaining corrected $p$-values > 0.05). **E**: The relationship between participants' confidence in their estimates for the features associated with suboptimal training policies and the proportion of test trials in which the more rewarding optimal training policy was selected. **F**: The same plot as E, except showing the participants' confidence in their estimates for the features associated with the optimal training policies. A, D: Error bars indicate standard error of the mean. B, C, E, F: Black lines indicate linear fits to the data and gray lines indicate 95% confidence intervals of the fits. A–F: Significance thresholds were corrected for 14 exploratory tests (5 pairwise comparisons for each bar plot and four correlations). The Bonferroni–Holm correction was used to correct for multiple comparisons. Materials to reproduce this figure are available at https://gin.g-node.org/sam.hall-mcmaster/sfgpi-behavioural-analysis.
(TIF)

**S7 Fig. Validation of the neural decoding pipeline.** To validate our decoding approach, we trained neural decoders to distinguish the city observed during feedback on training trials. We then tested how well we could recover the chosen city on held out training trials. One functional run was held out at a time to assess decoding performance. As training trials were not related to our hypotheses, we used the validation procedure to explore three possible time shifts (+4 s, +5 s, +6 s following feedback onset) and three possible smoothing levels (0 mm, 2 mm, 4 mm) that could be used to optimize decoder training. The validation results were based on 25 iterations. Data were subsampled at random on each iteration to match the number of trials from each class. The highest-performing decoding parameters were selected for each ROI based on the validation results. These parameters were later used when testing our hypotheses about neural reactivation during test tasks. **A**: Validation results for occipitotemporal cortex (OTC). The left subplot shows the mean decoding evidence for the selected city on held-out training trials for each combination of decoding parameters. The mean is taken across a 5 s window (3.75–8.75 s) following feedback onset. Bars indicate the standard error of the mean. The blue box denotes the decoding parameters with the highest validation performance. The right subplot shows the decoding time course based on those parameters. **B–D**: The remaining panels follow the same structure as A but present validation results for different brain regions. These include the medial temporal lobe (MTL) in B, the orbitofrontal cortex (OFC) in C, and the dorsolateral

prefrontal cortex (DLPFC) in D. A–D: Materials to reproduce this figure are available at https://gin.g-node.org/sam.hall-mcmaster/sfgpi-neural-analysis.
(TIF)

**S8 Fig. Comparison of cue-locked and feedback-locked decoding.** It is possible that the activation of city stimuli could occur directly following task cue onset. We therefore performed additional validation tests, examining the decoding evidence on training tasks when decoder training was locked to task cue onset, rather than feedback onset. Decoding based on feedback onset was found to be more effective in all ROIs. Panels **A–D** were generated using the same approach described in S7 Fig. For cue-locked decoding, the decoder was trained using data with a time shift of $+4\,s$, $+5\,s$, or $+6\,s$ from cue onset and evaluated from cue onset on held-out trials. Decoding evidence above represents evidence for the city selected on each training trial, averaged over a $5\,s$ window (3.75–8.75 s) following the relevant event onset (cue presentation or feedback presentation). **A**: Validation results for occipitotemporal cortex (OTC). Bars indicate the standard error of the mean. The blue box denotes the decoding parameters with the highest validation performance. **B–D**: The remaining panels follow the same structure as A but present validation results for different brain regions. These include the medial temporal lobe (MTL) in B, the orbitofrontal cortex (OFC) in C, and the dorsolateral prefrontal cortex (DLPFC) in D. A–D: Materials to reproduce this figure are available at https://gin.g-node.org/sam.hall-mcmaster/sfgpi-neural-analysis.
(TIF)

**S9 Fig. Decoding evidence for policies on each individual test task.** Each panel shows average decoding evidence during a specific test task (y-axis) as a function of brain region (x-axis). Colors denote different policy categories, and small translucent circles show data from individual participants. Panel **B** is especially relevant for testing whether the neural data are more consistent with SF&GPI or Universal Value Function Approximation (UVFA). The theoretical choice profiles for SF&GPI and UVFA are distinct for test task $w = [1,1,-1]$ (S4 Fig). The theoretical choice profile for UVFA predicts: 1) that the more rewarding training policy and objective best policy will be activated on test task $w = [1,1,-1]$ and 2) that the decoding evidence for these two policies will be comparable in magnitude. In contrast, we found that the more rewarding training policy was activated in OTC ($t(37) = 3.87$, corrected $p = 0.003$) but did not detect evidence that the objective best policy was activated ($t(37) = -0.90$, corrected $p = 0.750$). Decoding evidence for the more rewarding training policy was also significantly higher than the objective best policy ($t(37) = 3.21$, corrected $p = 0.014$). Equivalent tests in DLPFC were not significant (t-values < 1.66, corrected p-values > 0.094). These follow-up tests were restricted to OTC and DLPFC based on the decoding results in Fig 3 of the main text. p-values were corrected for six tests using the Bonferroni–Holm correction (2 policy categories × 2 ROIs and two follow-up tests assessing the difference in decoding evidence in each ROI). Materials to reproduce this figure are available at https://gin.g-node.org/sam.hall-mcmaster/sfgpi-neural-analysis.
(TIF)

**S10 Fig. Associative memory paradigm used to train the feature decoders.** On each trial, participants were shown a retrieval cue and need to select a corresponding target stimulus. Targets were either feature numbers or cities that would later appear during the gem collector game. Participants were pre-trained on the associations before scanning, and two retrieval cues were used per target stimulus. Feature decoders were then trained using neural patterns that arose from the response phase onset. For a full description, please see *Session One* and *Feature Decoding* in the Methods.
(TIF)

**S11 Fig. Validation tests of feature decoding during the associative memory paradigm.** To test the validity of our feature decoding approach, we trained neural decoders to distinguish the target number presented during the response phase. Response phase onset was used for training to be as consistent as possible with the policy decoding approach in S7 Fig, given the associative memory paradigm did not have feedback after correct responses. The validation tests used the optimal smoothing and time shift for each ROI identified in S7 Fig. Once the feature decoders were trained, we

tested how well we could recover the target number on held-out trials. One functional run was held out at a time to assess decoding performance. The process was repeated 25 times. Data were subsampled at random on each iteration to match the number of trials from each target class. **A**: Decoding time course for occipitotemporal cortex (OTC). Bars indicate the standard error of the mean. The dotted line indicates the chance based on 12 possible classes. **B–D**: The remaining panels follow the same structure as A but present validation results for different brain regions. These include the medial temporal lobe (MTL) in B, the orbitofrontal cortex (OFC) in C, and the dorsolateral prefrontal cortex (DLPFC) in D. The target number could be recovered in OTC, OFC, DLPFC but not MTL using this approach. Materials to reproduce this figure are available at https://gin.g-node.org/sam.hall-mcmaster/sfgpi-neural-analysis.
(TIF)

**S12 Fig. Decoding evidence for feature triplets on test tasks.** Another neural hypothesis formulated based on the SF&GPI algorithm is that feature triplets associated with the optimal training policies should be reactivated on test trials. This figure shows average decoding evidence for features associated with the more and less rewarding training policies on test trials (y-axis) as a function of brain region (x-axis). Feature information could not be decoded above chance in the four brain regions of interest (corrected $p$-values > 0.05). Materials to reproduce this figure are available at https://gin.g-node.org/sam.hall-mcmaster/sfgpi-neural-analysis.
(TIF)

**S13 Fig. Comparison of city decoders trained on session one and session two data.** Target cities could be recovered based on neural activity in all ROIs during the associative memory paradigm (session one). However, decoding evidence for cities during the gem collector paradigm (session two) was often higher when decoders were trained on the data from that same paradigm. **A**: Decoding evidence in OTC. The left subplot shows decoding evidence for target cities during the associative memory paradigm. The right subplot shows decoding evidence for cities selected on training trials of the gem collector paradigm, when the decoder was trained on data from session one (blue) or session two (orange). Bars indicate the standard error of the mean, and the dotted line indicates chance. **B–D**: Panels follow the same structure as A but present decoding time courses for different ROIs. A–D: Decoder training used the optimal smoothing and time shift for each ROI identified in S7 Fig, and 25 subsampling iterations were performed to match the number of trials from each target class. Decoding evidence presented above is cross-validated, based on assessment on held-out test data. Materials to reproduce this figure are available at https://gin.g-node.org/sam.hall-mcmaster/sfgpi-neural-analysis.
(TIF)

## Author contributions

**Conceptualization:** Sam Hall-McMaster, Momchil S. Tomov, Samuel J. Gershman, Nicolas W. Schuck.

**Data curation:** Sam Hall-McMaster.

**Formal analysis:** Sam Hall-McMaster.

**Funding acquisition:** Sam Hall-McMaster, Samuel J. Gershman, Nicolas W. Schuck.

**Investigation:** Sam Hall-McMaster.

**Methodology:** Sam Hall-McMaster, Momchil S. Tomov, Samuel J. Gershman, Nicolas W. Schuck.

**Resources:** Momchil S. Tomov.

**Supervision:** Momchil S. Tomov, Samuel J. Gershman, Nicolas W. Schuck.

**Visualization:** Sam Hall-McMaster.

**Writing – original draft:** Sam Hall-McMaster.

**Writing – review & editing:** Sam Hall-McMaster, Momchil S. Tomov, Samuel J. Gershman, Nicolas W. Schuck.

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
