## [Editor Report · Decision Letter 0]

4 Sep 2024

Dear Dr Hall-McMaster,

Thank you for submitting your manuscript entitled "Neural Prioritisation of Past Solutions Supports Generalisation" for consideration as a Research Article by PLOS Biology.

Your manuscript has now been evaluated by the PLOS Biology editorial staff and I am writing to let you know that we would like to send your submission out for external peer review.

Once your full submission is complete, your paper will undergo a series of checks in preparation for peer review. After your manuscript has passed the checks it will be sent out for review. To provide the metadata for your submission, please Login to Editorial Manager (https://www.editorialmanager.com/pbiology) within two working days, i.e. by Sep 06 2024 11:59PM.

Kind regards,

Christian

Christian Schnell, PhD

Senior Editor

PLOS Biology

cschnell@plos.org

---

## [Decision Letter · Decision Letter 1]

14 Jan 2025

Dear Sam,

Thank you for your patience while your manuscript "Neural Prioritisation of Past Solutions Supports Generalisation" was peer-reviewed at PLOS Biology. I'd like to apologize again for this very long delay. As I mentioned before, we had difficulties with multiple reviewers dropping out at different stages in the process, meaning I had to sign on new reviewers late in the process. In any case, your manuscript has now been evaluated by the PLOS Biology editors, an Academic Editor with relevant expertise, and by several independent reviewers.

In light of the reviews, which you will find at the end of this email, we would like to invite you to revise the work to thoroughly address the reviewers' reports.

As you will see below, the reviewers think that your study is well executed and provides important insights. Reviewer 1 requests only textual clarifications and additional discussions, while Reviewer 2 requests a deeper examination of both the behavioral and neural data to strengthen the study's claims and provide further support for the conclusions. We encourage you to revise your manuscript carefully in light of the reviewers' detailed suggestions.

Given the extent of revision needed, we cannot make a decision about publication until we have seen the revised manuscript and your response to the reviewers' comments. Your revised manuscript is likely to be sent for further evaluation by all or a subset of the reviewers.

**IMPORTANT - SUBMITTING YOUR REVISION**

*Re-submission Checklist*

*Published Peer Review*

*PLOS Data Policy*

*Blot and Gel Data Policy*

Sincerely,

Christian

Christian Schnell, PhD

Senior Editor

PLOS Biology

cschnell@plos.org

REVIEWS:

Reviewer #1: In the current paper, the authors investigate an interesting and timely question about the neural basis of a particular form of generalization, where human participants could re-use previous solutions in new problems. Behavior matched the predictions of their SF&GPI model, as previously behaviorally validated in an earlier publication. fMRI results focused on decoding representations of choice options, where the previously favored options were reactivated in new situations. Unexpectedly, the effect in the visual cortex was related to generalization behavior. The neural findings are interesting and novel. Overall, the paper was a pleasure to read. It has a very clear and comprehensive description of the experiment and results, and it is obvious that considerable care went into designing an interesting and well-controlled task and selecting appropriate methods. My concerns are primarily about interpretation, regarding a potential alternative account and the comparison to a model-based alternative.

1. The SF&GPI model is clear and it fits behavior well. However, I think it is important to describe how this model overlaps with or differs from an alternative perspective, based on configural / conjunctive learning, as highlighted by recent human imaging papers (e.g. Duncan et al. 2018 Neuron; Ballard et al. 2019 Nat Comm).

In the current task, one could construct an MF learning model that operates over configurations (cue screen displaying a +$ associated with a single gem), linking configurations to options (cities) based on reinforcement. This seems to be the same as a 'policy' in the SF&GPI model, but derived instead from an MF model based on conjunctions (and supported in implementation by the papers cited above).

For example, during training, consider the case where the cue screen configuration of [gem1 with +$ below] is followed by reward when city4 is chosen, while a cue screen configuration of [gem3 with +$ below] is followed by reward when city1 is chosen. Individual gems are not associated with reward, and individual cities are not associated with reward, but specifically the conjunction. Presenting that conjunction again will favor a response of the associated city, based on the stored values.

In the weather prediction task (e.g. Duncan et al.) this situation would be similar to a cue configuration of AB followed by reward when 'sun' is selected, while the configuration of AC or DB is followed by reward when 'rain' is selected.

In the current task, if these reinforced configuration-city links are carried over into the test trials, then the prediction is that participants would choose them instead of re-evaluating the alternatives. The behavioral and neural predictions from this configural account seem like they would be very similar or exactly the same as those of the SF&GPI account. The authors should explain the overlap with the SF&GPI model, and whether and how a configural account could explain the current data.

Then a minor note about the neural predictions - in the configural account and also I think in the SF&GPI model, there is no extra need to activate the features (gem numbers), because the favored training policy can be selected without retrieval of these specific details. In a configural account, it also isn't clear that reactivation of the options would be expected to occur at all - especially if the configuration is well-learned. Perhaps this will not happen in the current experiment, as training is short and there are effectively many 'reversals' of configural responses across all the blocks.

2. I have two concerns about the strong interpretation of the support for the SF&GPI account of behavior over a model-based account. It seems that an MB agent with some reasonable additions - weighing noisy memory and trading of incentives versus costs of computation - would provide very similar results as the SF&GPI account, and so perhaps the interpretation can be tempered a bit.

First, subjects' actual task experiences during training, as shown in the individual subject data points shown in Figure 2C, only rarely include the two non-favored cities. Because they only have rare experience of the features (gem numbers) in those cities, and then even when experienced, they receive negative feedback, memory for them will be noisy and inaccurate, relative to the two other cities. The current MB agent assumes perfect memory for features in all cities, but this produces a bit of a straw model for comparison.

In an MB agent that takes into account the strength of memory, when gem value estimates are noisy and inaccurate, then when faced with a new situation in a test trial, this agent would weigh options by the degree of expected estimation errors, and, if the weight on memory in the decision were high enough, the MB agent would be biased to choose the previously optimal options - mirroring the SF&GPI agent.

The memory strength issues is something that the authors obliquely address in the supplement, to their credit. In the last block only, participants reported final gem count estimates and confidence ratings for each gem in each city. As this was limited to the last block, it likely reduces the power to detect some effects of memory errors and confidence and optimal policy. But nevertheless, the authors find a strong correlation between memory errors and the use of the SF&GPI policy, the same policy that would also be favored by an MB agent with uncertain memory. In the behavioral results, it would help to point to the correlation between error in gem count estimates for options disfavored during training & policy re-use.

Second, and less importantly, an MB agent with reasonable constraints may weigh whether full re-evaluation is worth the effort costs. In the current case, the relative reward improvement at test between the previously favored option and the new optimal option may have been insufficient to overcome the cost of doing a full MB evaluation (perhaps on top of the consideration of noisy memory).

Specifically, at test, it seems like the earnings benefit is only yielding improvements of at most 12.5% (or 10-20 points, as noted in the discussion section) over the option preferred during training. In contrast, during training, the earnings benefit over the next-best option seems to be at least a 200% improvement (though some alternatives are all negative). To estimate the reward difference, of course, an MB agent would need to try this evaluation a few times in the first place, but once the minimal point gain in testing is checked in early blocks, the full evaluation could be dropped to save costs. (Adding this effort cost tradeoff involves some extra machinery than the simpler memory noise point.)

In the task description, I would suggest the authors add a note on the difference in the average delta between the optimal and next-best options during training and during test. (Based on quick calculations, that delta ranges from 60 to 180 points during training, but as described it is only 10-20 during test.) Of course, the authors were constrained in the experimental design by the arithmetic and conditions, so a greater benefit in test trials may not have been possible, but being clear about the values would help.

Taken together, these points about a more reasonable MB agent suggest that the current strong favoring of the SF&GPI agent be tempered a bit. These points do not affect the interpretation of the neural results.

3. In the behavior-to-neural correlation analysis shown in Fig. 3F, it isn't clear how much this result may be associated with the response made on those trials. Controlling for the response made is an extra analysis done in the previous section, but it is omitted here. Please explain.

Minor points:

- The number of subjects in the abstract isn't what went into the reported analysis, which seems to be 38.

- For classifier training, the authors report the mean number of trials that go into training, but what is the minimum (and maximum) for each of these analyses?

- Did the associative reactivation of points on day 1 lead to significant classification of the point features, when tested on day 1 alone?

- Did the associative reactivation of cities on day 1 work on day 1 data, and did it lead to a similar-performing classifier the one built from day 2 data?

- Regarding the effects of feature triplet 4, was this related to the relative point difference between the best and next-best training option? I wasn't sure whether the features listed in order in the methods actually mapped to the order in the relevant supplemental figure, so it was hard to check.

Reviewer #2: This study by Hall-McMaster and colleagues aims to test a few predictions about the neural substrates of generalization. Using a task called the gem collector task, the authors present behavioral evidence suggesting that when participants transferred prior knowledge of the environment to adapt to changing reward contingencies, their behavior aligned most closely with the successor features (SF)-generalized policy improvement (GPI) algorithm. By decoding fMRI data from several regions of interest (ROIs), they found that representations of optimal training policies remained stronger during test trials, even when those policies were no longer optimal. Furthermore, among the optimal training policies, the one that yielded higher rewards during test trials was more strongly activated.

Overall, the results are original and compelling, offering insights into the neural substrates of flexible learning in humans. The experimental design builds on a previous study by Tomov et al. (2021), and the data analyses consist of straightforward behavioral analyses and neural decoding. The statistical analyses appear sound and align with standard procedures in fMRI research (though we note that we are not experts in fMRI). The supplementary information and figures are helpful, and both the behavioral and neural data appear to be available online.

Although the findings are clearly presented, a deeper examination of both the behavioral and neural data could enhance their impact. Below, we present several questions and propose additional analyses that could further strengthen the study's claims and potentially uncover additional insights. Some of the proposed analyses are essential to fully support the current conclusions.

1. The overall behavioral results are clear and convincing. However, there are a few subtle but important behavioral effects that could benefit from more explanation. First is that there are significant differences between the proportion of option 1 and option 4 choices, which is not predicted by any of the models. The discussion section mentioned that a Universal Value Function Approximator (UVFA) could explain this discrepancy. It would be informative to include predictions from that model to demonstrate that it cannot account for the dominant patterns observed in the behavior, and potentially in the neural data, as effectively as SF-GPI. Additionally, some subtleties in participants' behavior are discussed in the discussion section. These would make valuable supplementary analyses for the main findings and could be integrated into the results section for greater clarity and impact.

2. Based on the choice proportions (Fig. 1C), it seems that participants did not use a strictly optimal policy during training, since the non-optimal options were chosen with some frequency. Did participants choose among the non-optimal options randomly (i.e., an epsilon-greedy policy) or did they favor any one of the non-optimal options in each task? Do individual differences in training trials predict their behavior in test trials (maybe explaining the subgroups in Fig. S2)? If so, this would provide even stronger evidence for transfer of training policies to test trials.

3. For the fMRI results, we wonder why TRs after feedback onset were used for training the decoder, since it seems sensible that signal related to upcoming city choices could appear as early as cue onset. However, in Fig. S6, it looks like the representation of chosen city only arose after feedback onset. Does this suggest that during choice, the representation of city is different from during feedback? If so, would one expect different results if the TRs around cue onset were used for training?

4. It is surprising that there is such a strong and robust effect in occipitotemporal cortex (OTC) but not frontal regions for this value-based task. In general, what is the implication for this result for the neural substrates/mechanisms of reinforcement learning? Where does the value information that modulates OTC activity come from, if not in OFC, DLPFC, or MTL? Could these regions represent some other task-related variables (maybe related to motor response)?

5. Since behavioral results suggest that participants are not simply performing SF&GPI, we wonder if the lack of activation and prioritization of optimal training policies in OFC and MTL reflect a true lack of encoding, or that these areas encode some other type of decision signals (like more model-based signals), so that on average no policy is favored more than other ones. Maybe looking at the decoding evidence on a task-by-task basis will resolve this.

Minor point: The caption for Fig. 3F is incorrectly labeled as Fig. 3D. Additionally, it would be helpful to label each panel in this figure individually for clarity.

---

## [Decision Letter · Decision Letter 2]

25 Mar 2025

Dear Dr Hall-McMaster,

Thank you for your patience while we considered your revised manuscript "Neural Prioritisation of Past Solutions Supports Generalisation" for publication as a Research Article at PLOS Biology. This revised version of your manuscript has been evaluated by the PLOS Biology editors, the Academic Editor and the original reviewers.

Based on the reviews and on our Academic Editor's assessment of your revision, we are likely to accept this manuscript for publication, provided you satisfactorily address the remaining points raised by the reviewers and the following data and other policy-related requests:

* We would like to suggest a different title to improve its accessibility for our broad audience: "Human choices on new tasks rely on reusing strategies that were successful in previous tasks"

* Please add the links to the funding agencies in the Financial Disclosure statement in the manuscript details.

* Please include the approval/license number from the institutional review board.

* Please include information in the Methods section whether the study has been conducted according to the principles expressed in the Declaration of Helsinki.

DATA POLICY:

Regardless of the method selected, please ensure that you provide the individual numerical values that underlie the summary data displayed in the following figure panels as they are essential for readers to assess your analysis and to reproduce it: 2 (all panels), 3BCHI, S1ABC, S3 (all panels), S4 (all panels), S5 (all panels), S6 (all panels), S7CD, S8 (all panels), S9 (all panels) and S12

* Please ensure that you are using best practice for statistical reporting and data presentation. These are our guidelines https://journals.plos.org/plosbiology/s/best-practices-in-research-reporting#loc-statistical-reporting and a useful resource on data presentation https://journals.plos.org/plosbiology/article?id=10.1371/journal.pbio.1002128

* CODE POLICY

We expect to receive your revised manuscript within two weeks.

*Published Peer Review History*

*Press*

Sincerely,

Christian

Christian Schnell, PhD

Senior Editor

cschnell@plos.org

PLOS Biology

Reviewer remarks:

Reviewer #1 (G. Elliott Wimmer): Thank you for your consideration of a model-based alternative with noisy memory; I believe those adjustments have significantly improved the paper.

Thank you also for elaborating on the SF&GPI model relative to a configural learning account. I miss-estimated what would be needed for test phase performance in a configural account, which would indeed involve some kind of generalization or additional combination of features during training, e.g., for the 4th test task [1, 1, -1], as listed in the methods.

Otherwise, a simple (but not at all formalized!) 'pick the highest combination that has been previously reinforced' kind of generalization would appear to solve 3 of 4 tests, given that three have a positive gem value that aligns with training (where there was only 1 positive value in the sets).

As an aside, it is interesting that the behavior for test [1, 1, -1], which has the only mismatch with a (not formalized) configural learning + generalization account, shows the lowest level of behavior consistent with the SF&GPI account. If a configural + generalization strategy was dominating in general, in this case, the lack of match could prompt the use of additional evaluation for this specific test case.

Reviewer #2 (Alireza Soltani): The authors have adequately addressed all of our concerns. We have no further comments.

---

## [Editor Report · Decision Letter 3]

24 Apr 2025

Dear Sam,

Thank you for the submission of your revised Research Article "Neural evidence that humans reuse strategies to solve new tasks" for publication in PLOS Biology. On behalf of my colleagues and the Academic Editor, Matthew Rushworth, I am pleased to say that we can in principle accept your manuscript for publication, provided you address any remaining formatting and reporting issues. These will be detailed in an email you should receive within 2-3 business days from our colleagues in the journal operations team; no action is required from you until then. Please note that we will not be able to formally accept your manuscript and schedule it for publication until you have completed any requested changes.

PRESS

Sincerely, 

Christian

Christian Schnell, PhD

Senior Editor

PLOS Biology

cschnell@plos.org